# MODALITY ALIGNMENT ACROSS TREES ON HETEROGENEOUS HYPERBOLIC MANIFOLDS

**Wei Wu**[1,2*]**, Xiaomeng Fan**[1,2*]**, Yuwei Wu**[1,2]**, Zhi Gao**[1,2†]**, Pengxiang Li**[1,2]**,
Yunde Jia**[2,1†]**, Mehrtash Harandi**[3]

[1] Beijing Key Laboratory of Intelligent Information Technology, School of Computer Science & Technology, Beijing Institute of Technology

[2] Guangdong Laboratory of Machine Perception and Intelligent Computing, Shenzhen MSU-BIT University

[3] Department of Electrical and Computer System Engineering, Monash University

`https://mcislab-manifold-learning.github.io/HypModalAlign/`

## ABSTRACT

Modality alignment is critical for vision-language models (VLMs) to effectively integrate information across modalities. However, existing methods extract hierarchical features from text while representing each image with a single feature, leading to asymmetric and suboptimal alignment. To address this, we propose Alignment across Trees, a method that constructs and aligns tree-like hierarchical features for both image and text modalities. Specifically, we introduce a semantic-aware visual feature extraction framework that applies a cross-attention mechanism to visual class tokens from intermediate Transformer layers, guided by textual cues to extract visual features with coarse-to-fine semantics. We then embed the feature trees of the two modalities into hyperbolic manifolds with distinct curvatures to effectively model their hierarchical structures. To align across the heterogeneous hyperbolic manifolds with different curvatures, we formulate a KL distance measure between distributions on heterogeneous manifolds, and learn an intermediate manifold for manifold alignment by minimizing the distance. We prove the existence and uniqueness of the optimal intermediate manifold. Experiments on taxonomic open-set classification tasks across multiple image datasets demonstrate that our method consistently outperforms strong baselines under few-shot and cross-domain settings.

## 1 INTRODUCTION

In vision-language models (VLMs), modality alignment aims to bridge the modality gap and enable the effective integration of information across different modalities (Liang et al., 2022; Zhang et al., 2025). In real-world scenarios, multimodal semantics are inherently hierarchical (Dhillon et al., 2002; Stevens et al., 2024). For example, in biology, the semantics of an organism follow a taxonomy of kingdom, phylum, class, order, family, genus, and species. To align such hierarchical semantics across modalities, existing methods extract hierarchical features from textual labels while representing images using only a single feature (Khattak et al., 2023b; Li et al., 2024). A single visual feature is inherently asymmetric to hierarchical textual features, namely, it fails to capture the complete textual information, thereby causing suboptimal alignment, as shown in Figure 1.

In this paper, we propose Alignment across Trees, a method that constructs hierarchical features from both images and texts, and aligns such tree-like features to enhance modality alignment. Inspired by the findings that intermediate Transformer layers encode coarse information (Chen et al., 2024), we exploit the class tokens from intermediate layers to construct hierarchical visual features. To this end, we need to overcome two challenges: (1) extracting hierarchical visual features that carry coarse-to-fine semantic information. (2) Textual features are relatively pure, whereas visual features encode more complex and diverse information such as background (Pal et al., 2025), resulting in distinct geometric structures that reside on heterogeneous manifolds. Alignment across such heterogeneous manifolds remains underexplored and challenging.

---

*Equal contribution. †Corresponding authors.

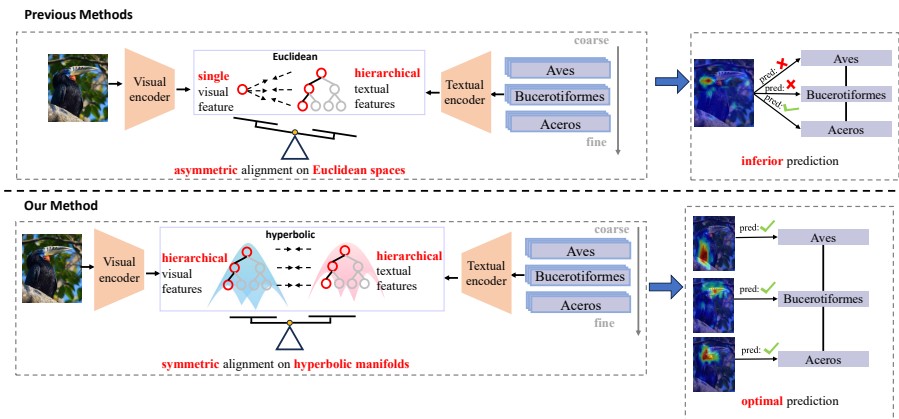

Figure 1: Comparison between previous methods and our method. Previous methods extract a single visual feature to align with hierarchical textual features in Euclidean spaces. This asymmetric alignment leads to inferior prediction. In contrast, our method achieves a symmetric alignment by extracting hierarchical visual features on hyperbolic manifolds, leading to optimal prediction.

To address the first challenge, we propose a semantic-aware framework that leverages textual cues to construct hierarchical visual features with coarse-to-fine semantics. Intermediate-layer tokens are projected into the final layer to enhance discriminative power. Next, we introduce a cross-attention module in which textual features at each hierarchy serve as queries and class tokens from different layers act as keys and values to produce hierarchy-specific visual features. The obtained visual features, paired with their textual counterparts, form symmetric textual–visual feature trees.

To address the second challenge, we introduce a heterogeneous manifold alignment algorithm. Hyperbolic manifolds, with their negative curvature, naturally model data hierarchies (Nickel & Kiela, 2017; Gao et al., 2022a). To capture geometric differences between textual and visual modalities, we embed each modality into an individual hyperbolic manifold with learnable curvature, allowing the feature trees to adapt to their intrinsic geometry. To align the two trees on heterogeneous hyperbolic manifolds, we explore an intermediate manifold close to both textual and visual manifolds. Specifically, we model data on each hyperbolic manifold as a wrapped normal distribution (Nagano et al., 2019; Gao et al., 2022b) and develop the KL divergence to measure manifold distance. By minimizing the derived distance between the intermediate manifold and textual and visual manifolds, we obtain the optimal intermediate manifold and prove its existence and uniqueness. We then use hyperbolic cones on the intermediate manifold to align cross-modal tree features while imposing geometric constraints on the visual and textual trees. Curvatures are optimized efficiently using the implicit function theorem (Lorraine et al., 2020) for curvature gradients.

We evaluate our approach on the taxonomic open-set (TOS) classification task (Wu et al., 2024) using four image datasets under four settings: few-shot, base-to-base, base-to-novel, and base-to-whole. We use three evaluation metrics (Wu et al., 2024): Leaf Accuracy (LA), Hierarchical Consistent Accuracy (HCA), and Mean Treecut Accuracy (MTA). Our method consistently surpasses all baselines across datasets and settings, showing clear advantages in modality alignment. Specifically, for HCA, our method improves by up to 7.72% (1-shot) and 28.83% (16-shot), highlighting its effectiveness. Moreover, visualizations confirm that the extracted image-feature trees capture hierarchical semantics, while ablation studies validate the contribution of the heterogeneous manifold alignment algorithm. Our contributions can be summarized as:

- We propose Alignment across Trees, a method that constructs hierarchical features from both images and texts, and aligns such tree-like features, thereby enhancing modality alignment.
- We introduce a semantic-aware visual feature extraction framework, which builds hierarchical visual features with coarse-to-fine semantics.
- We introduce a heterogeneous manifold alignment algorithm, which embeds image and text feature trees into hyperbolic manifolds with distinct curvatures and aligns them via an intermediate manifold optimized through manifold distance minimization.

## 2 RELATED WORKS

### 2.1 MODALITY ALIGNMENT

Existing modality alignment methods can be categorized into pre-training and prompt learning. Pre-training methods, such as CLIP (Radford et al., 2021) and ALIGN (Jia et al., 2021), train vision-language models (VLMs) on large-scale image-text pairs to bridge the modality gap. Recent efforts focus on scaling datasets (Gadre et al., 2024; Schuhmann et al., 2022) or improving training strategies (Li et al., 2023; Sun et al., 2023). Prompt learning introduces learnable prompt tokens at the input for modality alignment with significantly fewer computational resources than pre-training (Gu et al., 2023). Representative works include CoOp for continuous prompt optimization in the language branch (Zhou et al., 2022b), CoCoOp for conditional prompts based on visual features (Zhou et al., 2022a), and VPT for optimizing visual prompt tokens (Jia et al., 2022); subsequent studies explore multi-modal prompt fusion (Khattak et al., 2023a; Zhang et al., 2025), distribution estimation (Fan et al., 2025a), and regularization techniques (Zhu et al., 2023; Khattak et al., 2023b; Park et al., 2024). Some works use relative representations to preserve relational geometry, enabling communication between embedding spaces without enforcing a shared space. For example, Moschella et al. (2023) demonstrate that pairwise relational structure enables information transfer across embedding spaces, and Cannistraci et al. (2024) extend this principle through invariance-preserving transforms. Different from existing methods that model data on Euclidean space and use latent spaces, our method utilizes hyperbolic manifolds for modeling and constructs an intermediate manifold for communication.

Taxonomy classification predicts labels at multiple hierarchy levels. Several works (Goo et al., 2016; Kim & Frahm, 2018) utilize CNN-based networks to learn a hierarchical feature space. These methods rely on predefined hierarchical structures and fixed sets of categories for classification. To solve this issue, some works construct a multi-modal alignment method for taxonomy classification. Wu et al. (2024) first extract a single visual feature and compute contrastive losses with multi-level textual features using prompt learning, introducing metrics for hierarchical consistency. BioCLIP (Stevens et al., 2024), BioCLIP2 (Gu et al., 2025), and Biotrove (Yang et al., 2024) form prompts from coarse-to-fine annotations for pretraining. Sastry et al. (2025) apply transitive entailment constraints to features of hierarchical textual labels. Moreover, some works implicitly try to learn an identifiable hierarchical structure. For example, Kivva et al. (2022) study identifiability in deep generative models, and Kong et al. (2024) explore latent hierarchies by introducing structured discrete variables that capture hierarchical dependencies. However, these methods align hierarchical textual features with a single visual representation (*i.e.*, asymmetric alignment) on the Euclidean spaces, leading to suboptimal alignment and ignoring the geometric structure of hierarchical multi-modal data. In contrast, our method extracts hierarchical visual features to align hierarchical textual features (*i.e.*, symmetric alignment), and we model them in hyperbolic manifolds for improved alignment. The reason we choose hyperbolic manifolds is that they offer a more effective way to capture hierarchical geometric structures, as their volume grows exponentially with the radius, aligning with the exponential increase in data size along the depth of hierarchical structures (Fan et al., 2025b).

### 2.2 LEARNING ON HYPERBOLIC MANIFOLDS

Modeling via hyperbolic manifolds has shown superior performance in many tasks due to their capabilities in encoding data with hierarchical structures. Hyperbolic neural networks (Guo et al., 2022; Shimizu et al., 2021; He et al., 2025b; Malik et al., 2025; Skopek et al., 2020; Gao et al., 2021; Yu et al., 2025) incorporate several hyperbolic operations on top of a neural network to obtain hyperbolic embeddings. Recently, hyperbolic neural networks have been applied to diverse modalities such as graphs (Fu et al., 2023; 2024; Malik et al., 2025), text (He et al., 2025a), images (Wang et al., 2024b; Franco et al., 2024; Li et al., 2025b; Gao et al., 2023; Li et al., 2025a), videos (Long et al., 2020; Hong et al., 2023a), and audio (Hong et al., 2023b). Moreover, recent work (Ramasinghe et al., 2024; Desai et al., 2023; Pal et al., 2025; Wang et al., 2024a) has focused on developing multimodal methods on hyperbolic manifolds by combining entailment learning with CLIP to learn embeddings in hyperbolic manifolds. Mandica et al. (2024) explore hyperbolic embeddings in VLMs with billions of parameters. Existing methods assume that the curvatures of visual and textual modalities are the same, which cannot precisely model the geometric structures of each modality, further hindering the effectiveness. In contrast, we model visual and textual modal-

ities on manifolds with different curvatures for precisely capturing their geometries, and we design a heterogeneous manifold alignment algorithm for better alignment.

## 3 PRELIMINARIES

**Hyperbolic manifold.** Unlike Euclidean spaces with zero curvature, a hyperbolic manifold is a smooth Riemannian manifold with constant negative curvature $-c$ ($c > 0$) (Lee, 2006). We choose the Lorentz model $\mathcal{L}^c$ (Cannon et al., 1997) for hyperbolic manifolds due to its computational efficiency and numerical stability. The Lorentz model is defined as $\mathcal{L}^c = \{ \boldsymbol{x} \in \mathbb{R}^{n+1} : \langle \boldsymbol{x}, \boldsymbol{x} \rangle_{\mathcal{L}} = -\frac{1}{c} \}$. The hyperbolic vector can be written as $\boldsymbol{x} = [\boldsymbol{x}_{space}, x_{time}]$, where $x_{time} \in \mathbb{R}$ corresponds to the hyperboloid's axis of symmetry and $\boldsymbol{x}_{space} \in \mathbb{R}^n$ represents the remaining spatial coordinates. $\langle \cdot, \cdot \rangle_{\mathcal{L}}$ denotes the Lorentzian inner product that is computed as $\langle \boldsymbol{x}, \boldsymbol{y} \rangle_{\mathcal{L}} = \langle \boldsymbol{x}_{space}, \boldsymbol{y}_{space} \rangle - x_{time} y_{time}$, where $\langle \cdot, \cdot \rangle$ is the Euclidean inner product. The induced Lorentzian norm is $\|\boldsymbol{x}\|_{\mathcal{L}} = \sqrt{|\langle \boldsymbol{x}, \boldsymbol{x} \rangle_{\mathcal{L}}|}$. The following hyperbolic operations are used in our work.

**Distance.** The Lorentzian distance between $\boldsymbol{x}, \boldsymbol{y} \in \mathcal{L}^c$ is $d_{\mathcal{L}}(\boldsymbol{x}, \boldsymbol{y}) = \sqrt{1/c} \cdot \mathrm{arccosh}\left(-c \langle \boldsymbol{x}, \boldsymbol{y} \rangle_{\mathcal{L}}\right)$.

**Tangent space.** The tangent space to $\mathcal{L}^c$ at a tangent point $\boldsymbol{x}$, denoted as $\boldsymbol{T}_{\boldsymbol{x}}\mathcal{L}^c$, consists of all tangent vectors at that tangent point. Any vector in ambient space $\boldsymbol{u} \in \mathbb{R}^{n+1}$ can be projected to the tangent space $\boldsymbol{T}_{\boldsymbol{x}}\mathcal{L}^c$ via $\boldsymbol{v} = \mathrm{proj}_{\boldsymbol{x}}^c(\boldsymbol{u}) = \boldsymbol{u} + c\boldsymbol{x}\langle \boldsymbol{x}, \boldsymbol{u} \rangle_{\mathcal{L}}$.

**Exponential map.** The exponential map $\mathrm{expm}_{\boldsymbol{x}}^c(\boldsymbol{v})$ projects $\boldsymbol{v}$ from $\boldsymbol{T}_{\boldsymbol{x}}\mathcal{L}^c$ to $\mathcal{L}^c$ as

$$\mathrm{expm}_{\boldsymbol{x}}^c(\boldsymbol{v}) = \cosh\left(\sqrt{c}\|\boldsymbol{v}\|_{\mathcal{L}}\right) \boldsymbol{x} + \frac{\sinh\left(\sqrt{c}\|\boldsymbol{v}\|_{\mathcal{L}}\right)}{\sqrt{c}\|\boldsymbol{v}\|_{\mathcal{L}}} \boldsymbol{v}. \tag{1}$$

**Logarithmic map.** The logarithmic map $\mathrm{logm}_{\boldsymbol{y}}^c(\boldsymbol{x})$ projects a vector $\boldsymbol{x}$ from $\mathcal{L}^c$ to $\boldsymbol{T}_{\boldsymbol{y}}\mathcal{L}^c$ as

$$\mathrm{logm}_{\boldsymbol{y}}^c(\boldsymbol{x}) = \frac{\mathrm{arccosh}(-c\langle \boldsymbol{y}, \boldsymbol{x} \rangle_{\mathcal{L}})}{\sqrt{(c\langle \boldsymbol{y}, \boldsymbol{x} \rangle_{\mathcal{L}})^2 - 1}} \mathrm{proj}_{\boldsymbol{y}}^c(\boldsymbol{x}). \tag{2}$$

**Wrapped normal distributions.** The wrapped normal distribution (Nagano et al., 2019) on $\mathcal{L}^c$ is defined as

$$\mathcal{N}_{\mathcal{L}^c}(\boldsymbol{x} \mid \boldsymbol{u}, \delta) = \frac{1}{Z(\delta)} \exp\left(-\frac{d_c^2(\boldsymbol{x}, \boldsymbol{u})}{2\delta^2}\right), \tag{3}$$

where $\boldsymbol{u}$ is the Fréchet mean, $\delta > 0$ is a dispersion parameter, and $Z(\delta)$ is a dispersion dependent normalization constant.

**Hyperbolic entailment cones.** Hyperbolic entailment cones $\omega(\boldsymbol{x})$ are regions for every possible point $\boldsymbol{x}$ in the hyperbolic manifold, such that all points $\boldsymbol{y} \in \omega(\boldsymbol{x})$ are semantically linked to $\boldsymbol{x}$ as its child concepts. As such, points in $\omega(\boldsymbol{x})$ are expected to contain specific information for the general concept $\boldsymbol{x}$. The cone is defined by half-aperture

$$\omega(\boldsymbol{x}) = \sin^{-1}\left(\frac{2k}{\sqrt{c}\|\boldsymbol{x}_{space}\|}\right), \tag{4}$$

where $k = 0.1$ is a constant.

## 4 METHOD

We propose the method, Alignment across Trees, which contains a semantic-aware visual feature extraction framework to construct textual and visual feature trees and a heterogeneous manifold alignment algorithm to align the feature trees (See Figure 2).

### 4.1 SEMANTIC-AWARE VISUAL FEATURE EXTRACTION FRAMEWORK

Existing vision-language models (*e.g.*, CLIP and BLIP) and prompt learning methods (*e.g.*, CoOp) typically align tokens from the last layer of the Vision Transformers (ViT) with the textual modality. Recent studies show that intermediate layers of ViT encode coarse semantics, while the final layer encodes fine-grained information (Chen et al., 2024). Motivated by this observation, we leverage

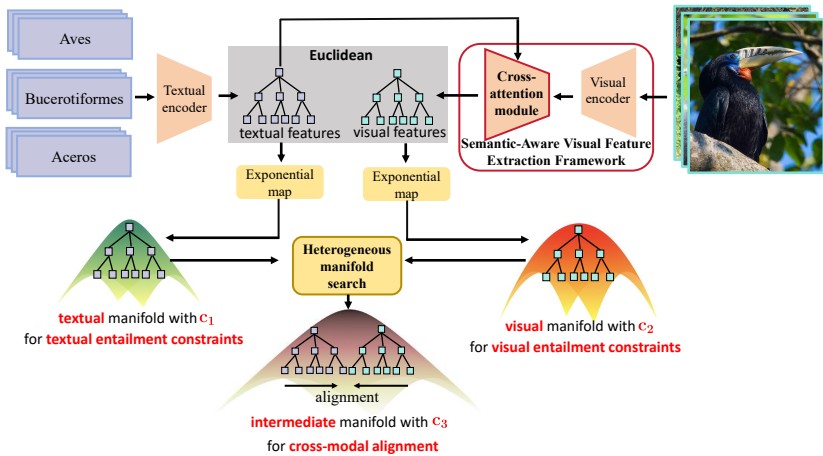

Figure 2: Pipeline of our method.

class tokens from $m$ intermediate layers $\{\boldsymbol{h}_{p_j}\}_{j=1}^m$ and the final layer (denoted by $\boldsymbol{h}_n$) to construct hierarchical visual features, as illustrated in Figure 3. We enhance the discriminative power of intermediate-layer tokens by mapping them to the final layer's representation space. Specifically, for $\boldsymbol{h}_{p_j}$, from the $p_j$-th layer onward, we disable cross-token self-attention by removing query and key computations so that each token no longer attends to others. $\boldsymbol{h}_{p_j}$ is then forwarded to the final layer through only linear projection, residual connections, and MLP updates, preserving its original information for alignment. The mapped tokens are denoted by $\boldsymbol{h}'_{p_j}$ ($\boldsymbol{h}_n$ doesn't need to be mapped).

To construct hierarchical visual features that align with hierarchical textual features $\{\boldsymbol{t}_i\}_{i=1}^H$, where $H$ denotes the depth of the textual hierarchy, we design a cross-attention mechanism where textual features serve as queries and class tokens from different layers ($\{\boldsymbol{h}'_{p_j}\}_{j=1}^m$ and $h_n$) serve as keys and values. The hierarchical visual features $[\boldsymbol{v}_1; \boldsymbol{v}_2; \ldots; \boldsymbol{v}_H]$ are computed as:

$$[\boldsymbol{v}_1; \boldsymbol{v}_2; \ldots; \boldsymbol{v}_H] = \mathrm{Softmax}\left(\frac{\boldsymbol{Q}\boldsymbol{K}^\top}{\sqrt{d}}\right)\boldsymbol{V}_{\mathrm{attn}},$$

$$\text{where} \quad \boldsymbol{Q} = [\boldsymbol{t}_1; \ldots; \boldsymbol{t}_H]\boldsymbol{W}_Q, \quad \boldsymbol{K} = [\boldsymbol{h}'_{p_1}; \ldots; \boldsymbol{h}_n]\boldsymbol{W}_K, \quad \boldsymbol{V}_{\mathrm{attn}} = [\boldsymbol{h}'_{p_1}; \ldots; \boldsymbol{h}_n]\boldsymbol{W}_V. \tag{5}$$

$\boldsymbol{W}_Q, \boldsymbol{W}_K, \boldsymbol{W}_V \in \mathbb{R}^{d \times d}$ are learnable parameters. In this way, we can model the hierarchical text feature tree $T_e = \{\boldsymbol{t}_i\}_{i=1}^H$ and hierarchical image feature tree $V_e = \{\boldsymbol{v}_i\}_{i=1}^H$, with the symmetric semantic information.

## 4.2 HETEROGENEOUS MANIFOLD ALIGNMENT ALGORITHM

The hyperbolic manifolds are well-suited for modeling hierarchical features. Given the geometric differences between textual and visual feature trees, we embed them in separate hyperbolic manifolds with distinct, learnable curvatures $c_1$ (text) and $c_2$ (image). Formally,

$$\boldsymbol{t}_i^{c_1} = \mathrm{expm}_{\boldsymbol{0}}^{c_1}(\boldsymbol{t}_i), \quad \boldsymbol{v}_i^{c_2} = \mathrm{expm}_{\boldsymbol{0}}^{c_2}(\boldsymbol{v}_i). \tag{6}$$

The curvatures $c_1$ and $c_2$ are data-driven and treated as trainable parameters optimized together with the model via the loss functions. To align the image features and text features located in different hyperbolic manifolds, the heterogeneous manifold alignment algorithm constructs an intermediate hyperbolic manifold, as shown in Figure 2.

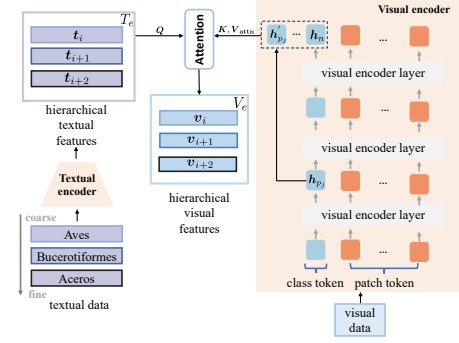

Figure 3: Structure of semantic-aware visual feature extraction framework.

### 4.2.1 INTERMEDIATE MANIFOLD CONSTRUCTION

To minimize geometric distortion and preserve the original structures, we introduce an intermediate manifold $\mathcal{L}^{c_3}$ that bridges the textual and visual manifolds. Directly measuring the dissimilarity between hyperbolic manifolds is underexplored. We define a distance function $D_{\mathcal{L}}(\cdot, \cdot)$ to quantify how dissimilar two hyperbolic manifolds are. Using this distance, the optimal curvature $c_3^*$ is obtained by optimizing the following objective,

$$c_3^* = \arg\min_{c_3} J_c(c_3) := D_{\mathcal{L}}(\mathcal{L}^{c_1}, \mathcal{L}^{c_3}) + D_{\mathcal{L}}(\mathcal{L}^{c_2}, \mathcal{L}^{c_3}). \tag{7}$$

We model the distributions on hyperbolic manifolds by wrapped normal distributions and use the KL divergence between the distributions to define the manifold distance $D_{\mathcal{L}}(\cdot, \cdot)$. Since the KL divergence on hyperbolic manifolds does not have an analytic expression (Cho et al., 2023), we present an approximate expression for the KL divergence and utilize it to define the manifold distance, as shown in Theorem 1.

**Theorem 1** *Given two manifolds $\mathcal{L}^{c_1}$ and $\mathcal{L}^{c_3}$, the distributions on the two manifolds are*

$$P_{c_1, \boldsymbol{u}_1} = \mathcal{N}_{\mathcal{L}}(\boldsymbol{x} \mid \boldsymbol{u}_1, \delta) = \frac{1}{Z(\delta)} e^{-\frac{d_{c_1}^2(\boldsymbol{x}, \boldsymbol{u}_1)}{2\delta^2}}, P_{c_3, \boldsymbol{u}_3} = \mathcal{N}_{\mathcal{L}}(\boldsymbol{x} \mid \boldsymbol{u}_3, \delta) = \frac{1}{Z(\delta)} e^{-\frac{d_{c_3}^2(\boldsymbol{x}, \boldsymbol{u}_3)}{2\delta^2}}, \quad (8)$$

*where $u_1$ and $u_3$ are the Fréchet means, and $\delta > 0$ is a dispersion parameter. We define the distance between $\mathcal{L}^{c_1}$ and $\mathcal{L}^{c_3}$ as an affine transformation of the Kullback-Leibler (KL) divergence, which is*

$$D_{\mathcal{L}}(\mathcal{L}^{c_1}, \mathcal{L}^{c_3}) = \frac{-\sqrt{c_1} + 2\sqrt{c_3} \cosh[(\sqrt{c_3} - \sqrt{c_1})r]}{2\sqrt{c_1}c_3}, \tag{9}$$

*where $r$ is a constant.*

Note that $D_{\mathcal{L}}(\cdot, \cdot)$ is not a formal metric due to the asymmetry and violation of the triangle inequality inherited from the KL divergence. We present the minimizer of $D_{\mathcal{L}}(\mathcal{L}^{c_1}, \mathcal{L}^{c_3})$ in Proposition 1 to show the soundness of $D_{\mathcal{L}}(\cdot, \cdot)$.

**Proposition 1** *The minimum of $D_{\mathcal{L}}(\mathcal{L}^{c_1}, \mathcal{L}^{c_3})$ in Eq.(9) is uniquely attained at $c_3 = c_1$.*

Proposition 1 shows that in Theorem 1, $\mathcal{L}^{c_1}$ and $\mathcal{L}^{c_3}$ are optimally aligned when $c_3 = c_1$, showing the soundness of Theorem 1. Based on the derived distance, $J_c(c_3)$ in Eq.(7) is formulated as

$$
\begin{aligned}
J_c(c_3) &= D_{\mathcal{L}}(\mathcal{L}^{c_1}, \mathcal{L}^{c_3}) + D_{\mathcal{L}}(\mathcal{L}^{c_2}, \mathcal{L}^{c_3}) \\
&= \frac{-\sqrt{c_1} + 2\sqrt{c_3} \cosh[(\sqrt{c_3} - \sqrt{c_1})r]}{2\sqrt{c_1}c_3} + \frac{-\sqrt{c_2} + 2\sqrt{c_3} \cosh[(\sqrt{c_3} - \sqrt{c_2})r]}{2\sqrt{c_2}c_3}.
\end{aligned} \tag{10}
$$

We demonstrate the existence and uniqueness of the minimizer of $J_c(c_3)$ by Proposition 2

**Proposition 2** *$J_c(c_3)$ has a unique minimizer $c_3^* \in \big[ \min\{c_1, c_2\}, \ \max\{c_1, c_2\} \big]$.*

Proofs of Theorem 1, Proposition 1, and Proposition 2 are provided in Appendix A. We apply the golden section search (Kiefer, 1953) to find the minimizer of $J_c(c_3)$, which can effectively solve the one-dimensional unconstrained optimization problem.

### 4.2.2 INTER-MODAL GEOMETRIC ALIGNMENT MECHANISM

After obtaining the curvature $c_3$, we utilize the exponential map to project the textual and visual features to $\mathcal{L}^{c_3}$,

$$\boldsymbol{t}_i^{c_3} = \mathrm{expm}_{\boldsymbol{0}}^{c_3}(\boldsymbol{t}_i), \quad \boldsymbol{v}_i^{c_3} = \mathrm{expm}_{\boldsymbol{0}}^{c_3}(\boldsymbol{v}_i). \tag{11}$$

Following Desai et al. (2023), we utilize the entailment to achieve inter-modal geometric alignment. Pal et al. (2025) shows that the text generally provides a broader context than images. Thus, as to each hierarchy, we force the visual feature $\boldsymbol{v}_i^{c_3}$ to be entailed in the textual feature $\boldsymbol{t}_i^{c_3}$, *i.e.*, $\boldsymbol{v}_i^{c_3}$ located

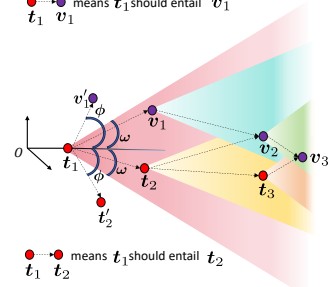

Figure 4: Illustration of entailment.

in $\omega(\boldsymbol{t}_i^{c_3})$ that is the entailment cone of $\boldsymbol{t}_i^{c_3}$. The loss function forces $\boldsymbol{v}_i^{c_3}$ to be in $\omega(\boldsymbol{t}_i^{c_3})$, which is modeled as

$$J_{ent}(\boldsymbol{v}_i^{c_3}, \boldsymbol{t}_i^{c_3}) = \max(0, \phi(\boldsymbol{v}_i^{c_3}, \boldsymbol{t}_i^{c_3}) - \omega(\boldsymbol{t}_i^{c_3})), \tag{12}$$

where $\phi(\boldsymbol{v}_i^{c_3}, \boldsymbol{t}_i^{c_3})$ is the exterior angle,

$$\phi(\boldsymbol{v}_i^{c_3}, \boldsymbol{t}_i^{c_3}) = \pi - \angle O\boldsymbol{t}_i^{c_3}\boldsymbol{v}_i^{c_3} = \cos^{-1}\left(\frac{v_i^{c_3}{}_{\text{time}} + t_i^{c_3}{}_{\text{time}} c\langle \boldsymbol{t}_i^{c_3}, \boldsymbol{v}_i^{c_3}\rangle_{\mathcal{L}}}{\|\boldsymbol{t}_i^{c_3}{}_{\text{space}}\|\sqrt{(c_3\langle \boldsymbol{t}_i^{c_3}, \boldsymbol{v}_i^{c_3}\rangle_{\mathcal{L}})^2 - 1}}\right). \tag{13}$$

Overall, for all hierarchies, the loss function for cross-modal alignment is modeled as

$$J_{ent}(V^{c_3}, T^{c_3}) = \sum_{i=1}^{H} J_{ent}(\boldsymbol{v}_i^{c_3}, \boldsymbol{t}_i^{c_3}) \tag{14}$$

where $V^{c_3} = \{\boldsymbol{v}_i^{c_3}\}_{i=1}^{H}$ and $T^{c_3} = \{\boldsymbol{t}_i^{c_3}\}_{i=1}^{H}$. The process is illustrated in Figure 4.

### 4.3 IN-MODAL GEOMETRIC STRUCTURE CONSTRAINTS

Since text and image features are hierarchical, we impose in-modal geometric structure constraints on each modality, requiring fine-grained features to be entailed by coarse-grained ones. We use the hyperbolic cones on $\mathcal{L}^{c_1}$ and $\mathcal{L}^{c_2}$ to model the textual entailment and visual entailment, respectively. For the $i$-th level (coarser) and the $i+1$-th level (finer) that are adjacent, we push $\boldsymbol{t}_{i+1}^{c_1}$ into cone $\omega(\boldsymbol{t}_i^{c_1})$, and $\boldsymbol{v}_{i+1}^{c_2}$ into cone $\omega(\boldsymbol{v}_i^{c_2})$. The losses on visual and textual modalities are formulated as

$$\begin{aligned} J_{ent}(\boldsymbol{v}_{i+1}^{c_2}, \boldsymbol{v}_i^{c_2}) &= \max(0, \phi(\boldsymbol{v}_{i+1}^{c_2}, \boldsymbol{v}_i^{c_2}) - \omega(\boldsymbol{v}_i^{c_2})), \\ J_{ent}(\boldsymbol{t}_{i+1}^{c_1}, \boldsymbol{t}_i^{c_1}) &= \max(0, \phi(\boldsymbol{t}_{i+1}^{c_1}, \boldsymbol{t}_i^{c_1}) - \omega(\boldsymbol{t}_i^{c_1})). \end{aligned} \tag{15}$$

For all hierarchies, the loss functions of hierarchical constraints on visual and textual modalities are

$$J_{Vent}(V^{c_2}) = \sum_{i=1}^{H-1} J_{ent}(\boldsymbol{v}_{i+1}^{c_2}, \boldsymbol{v}_i^{c_2}), \quad J_{Tent}(T^{c_1}) = \sum_{i=1}^{H-1} J_{ent}(\boldsymbol{t}_{i+1}^{c_1}, \boldsymbol{t}_i^{c_1}). \tag{16}$$

This process is also illustrated in Figure 4.

### 4.4 OPTIMIZATION STRATEGY

We employ CLIP to extract the hierarchical textual and visual features $T_e$ and $V_e$. Following the prompt learning paradigm, we introduce learnable tokens $\boldsymbol{\theta}$ to train our model. The overall loss function is formulated as

$$\begin{aligned} J(\boldsymbol{\theta}, c_1, c_2) &= J_{\text{pro}}(T_e, V_e) + \alpha\left(J_{\text{Tent}}(T^{c_1}) + J_{\text{Vent}}(V^{c_2}) + J_{ent}(V^{c_3^*}, T^{c_3^*})\right), \\ \text{s.t.} \quad c_3^* &= \arg\min_{c_3} J_c(c_3) := D_{\mathcal{L}}(\mathcal{L}^{c_1}, \mathcal{L}^{c_3}) + D_{\mathcal{L}}(\mathcal{L}^{c_2}, \mathcal{L}^{c_3}), \end{aligned} \tag{17}$$

where $J_{pro}$ is the loss in (Wu et al., 2024) and $\alpha$ is a weighting factor. We optimize the parameters using gradient descent,

$$\boldsymbol{\theta} \leftarrow \boldsymbol{\theta} - \eta \cdot \frac{dJ}{d\boldsymbol{\theta}}, \quad c_1 \leftarrow c_1 - \eta \cdot \frac{dJ}{dc_1}, \quad c_2 \leftarrow c_2 - \eta \cdot \frac{dJ}{dc_2}, \tag{18}$$

where $\eta$ denotes the learning rate, and the gradients with respect to $c_1$ and $c_2$ are computed as

$$\frac{dJ}{dc_1} = \frac{\partial J}{\partial c_1} + \frac{\partial J}{\partial c_3^*}\frac{\partial c_3^*}{\partial c_1}, \quad \frac{dJ}{dc_2} = \frac{\partial J}{\partial c_2} + \frac{\partial J}{\partial c_3^*}\frac{\partial c_3^*}{\partial c_2}. \tag{19}$$

However, the terms $\frac{\partial c_3^*}{\partial c_1}$ and $\frac{\partial c_3^*}{\partial c_2}$ cannot be computed through standard backpropagation due to the non-differentiable nature of the golden section search. To address this challenge, we apply the implicit function theorem to compute these derivatives,

$$\frac{\partial c_3^*}{\partial c_1} = -\left(\frac{\partial^2 J_c}{\partial c_3^2}\right)^{-1}\frac{\partial^2 J_c}{\partial c_1 \partial c_3}\bigg|_{c_3 = c_3^*}, \quad \frac{\partial c_3^*}{\partial c_2} = -\left(\frac{\partial^2 J_c}{\partial c_3^2}\right)^{-1}\frac{\partial^2 J_c}{\partial c_2 \partial c_3}\bigg|_{c_3 = c_3^*}. \tag{20}$$

# 5 EXPERIMENTS

## 5.1 SETTINGS

**Taxonomic Open Set (TOS) Classification.** TOS classification (Wu et al., 2024) organizes labels as a semantic tree, requiring the classifier to predict across multiple levels of semantics.

**Datasets.** We evaluate all methods on four datasets: Cifar100 (Krizhevsky et al., 2009), SUN (Xiao et al., 2010), ImageNet (Deng et al., 2009), and Rare Species (Stevens et al., 2024).

**Tasks.** We evaluate our method on two types of tasks. **(1) Few-shot.** We adopt a 1-shot and 16-shot training setting, where a fixed number of samples are randomly selected from each class. **(2) Base-to-base/base-to-novel/base-to-whole generalization.** We equally split each dataset into base and novel classes. The model is trained on base classes and evaluated on base classes (base-to-base), novel classes (base-to-novel), and whole classes (base-to-whole) across all 4 datasets.

**Metrics.** Following Wu et al. (2024), we evaluate performance using three metrics: Leaf Accuracy (LA), Hierarchical Consistent Accuracy (HCA), and Mean Treecut Accuracy (MTA).

**Implementation Details.** Following Wu et al. (2024), we conduct experiments based on the prompt tuning method MaPLe (Khattak et al., 2023a). We also conduct experiments on the PromptSRC (Khattak et al., 2023b) method to further validate the effectiveness of our method. More details of datasets, tasks, metrics, and implementation are provided in Appendix B.

## 5.2 FEW-SHOT SETTING RESULTS

Table 1: TOS classification results on the 1-shot and 16-shot settings. We bold the best results.

| K-Shot | Base Method | Variant | Cifar100 | | | SUN | | | ImageNet | | | Rare Species | | |
|---|---|---|---|---|---|---|---|---|---|---|---|---|---|---|
| | | | LA | HCA | MTA | LA | HCA | MTA | LA | HCA | MTA | LA | HCA | MTA |
| 1 | MaPLe | Vanilla | 68.75 | 4.65 | 50.60 | 63.98 | 25.15 | 50.31 | **68.91** | 2.97 | 48.16 | 41.55 | 5.09 | 44.75 |
| | | +ProTeCt | 69.33 | 48.10 | 83.36 | 64.29 | 50.45 | 76.73 | 66.16 | 20.44 | 85.18 | 39.92 | 13.22 | 70.04 |
| | | +Ours | **71.37** | **53.19** | **85.29** | 67.57 | **57.92** | **80.55** | 66.33 | **25.56** | **85.98** | **46.77** | **20.94** | **76.83** |
| | PromptSRC | Vanilla | 72.48 | 14.36 | 51.91 | 70.58 | 42.14 | 57.19 | 68.82 | 4.46 | 54.10 | 45.39 | 6.72 | 44.72 |
| | | +ProTeCt | 73.07 | 49.54 | 85.16 | 70.61 | 55.52 | 78.73 | 68.43 | 21.58 | 85.63 | 44.56 | 20.36 | 74.42 |
| | | +Ours | **73.54** | **51.91** | **85.76** | **70.64** | **57.79** | **79.94** | **68.86** | **25.13** | **86.45** | **46.98** | **23.03** | **77.32** |
| 16 | MaPLe | Vanilla | 75.01 | 17.54 | 52.21 | 71.86 | 33.25 | 54.29 | 70.70 | 4.15 | 48.16 | 50.94 | 5.30 | 40.41 |
| | | +ProTeCt | 75.34 | 61.15 | 88.04 | 72.17 | 59.71 | 82.27 | 69.52 | 31.24 | 87.87 | 48.14 | 24.82 | 78.79 |
| | | +Ours | **77.92** | **69.38** | **90.89** | **75.47** | **68.67** | **86.02** | **71.41** | **43.79** | **88.78** | **69.96** | **53.65** | **87.27** |
| | PromptSRC | Vanilla | 77.71 | 15.07 | 56.86 | 75.75 | 45.23 | 59.42 | 71.50 | 2.48 | 46.71 | 59.20 | 11.64 | 55.82 |
| | | +ProTeCt | 78.76 | 66.74 | 90.79 | 75.54 | 66.01 | 84.75 | 70.98 | 32.89 | 88.31 | 56.40 | 33.92 | 82.47 |
| | | +Ours | **78.90** | **68.47** | **91.12** | **76.54** | **69.18** | **86.20** | **71.67** | **42.26** | **89.64** | **67.38** | **50.77** | **87.60** |

We compare our method with MaPLe, PromptSRC, MaPLe + ProTeCt, and PromptSRC + ProTeCt methods in Table 1. Results show that our method significantly improves the performance of the compared methods in both 1-shot and 16-shot settings, demonstrating its effectiveness in modality alignment. Notably, in the 16-shot setting, our method achieves up to a $19.02\%$ improvement on LA, a $28.83\%$ improvement on HCA, and a $8.48\%$ improvement on MTA, indicating its ability to align the textual and visual modalities with the hierarchical semantic structures. See Appendix C.1 for comparisons between our methods and other hyperbolic alignment methods.

## 5.3 BASE-TO-BASE/BASE-TO-NOVEL/BASE-TO-WHOLE GENERALIZATION

Table 2 presents the comparison between our method and existing methods. Our method consistently outperforms all baselines across all metrics and settings, demonstrating strong generalization to novel classes. On Cifar100, our method achieves improvements of 1.38%, 5.66%, and 4.90% in LA, HCA, and MTA, respectively, on novel classes, highlighting the effectiveness of our method.

## 5.4 ABLATION STUDY

**Variants of our method.** To evaluate the effectiveness of each component, we design three variants: (i) **Ours-Euc**, which does not employ hyperbolic constraints; (ii) **Ours-HypV1**, where all

Table 2: Base-to-base/base-to-novel/base-to-whole generalization results across multiple datasets. See Appendix C.2 for more results.

| Dataset | Base Method | Variant | LA | | | | HCA | | | | MTA | | | |
|---|---|---|---|---|---|---|---|---|---|---|---|---|---|---|
| | | | Base | Novel | HM | Whole | Base | Novel | HM | Whole | Base | Novel | HM | Whole |
| SUN | MaPLe | Vanilla | 80.77 | 76.85 | 78.76 | 69.45 | 38.51 | 37.62 | 38.06 | 33.31 | 65.23 | 61.61 | 63.37 | 55.82 |
| | | +ProTeCt | 81.77 | 76.67 | 79.14 | 69.80 | 64.27 | 55.43 | 59.52 | 53.74 | 85.30 | 81.10 | 83.15 | 76.37 |
| | | +Ours | **82.79** | **77.11** | **79.85** | **69.82** | **73.38** | **56.23** | **63.67** | **57.09** | **88.85** | **81.24** | **84.87** | **78.61** |
| | PromptSRC | Vanilla | 82.30 | 78.68 | 80.58 | 71.50 | 51.77 | 48.25 | 49.94 | 47.05 | 68.89 | 65.67 | 67.24 | 58.93 |
| | | +ProTeCt | 82.36 | 78.40 | 80.33 | 71.94 | 66.86 | 58.80 | 62.57 | 57.06 | 86.67 | 82.76 | 84.67 | 79.12 |
| | | +Ours | **83.40** | **78.72** | **80.99** | **72.04** | **73.11** | **59.10** | **65.36** | **58.42** | **89.02** | **82.86** | **85.83** | **79.74** |
| Cifar100 | MaPLe | Vanilla | 82.60 | 75.80 | 79.05 | 71.45 | 9.94 | 6.94 | 8.17 | 7.00 | 61.46 | 50.66 | 55.54 | 54.14 |
| | | +ProTeCt | 82.66 | 74.56 | 78.40 | 70.03 | 61.84 | 35.86 | 45.40 | 39.37 | 89.65 | 77.15 | 82.93 | 77.34 |
| | | +Ours | **82.76** | **75.94** | **79.20** | **72.52** | **66.42** | **41.52** | **51.10** | **45.79** | **91.01** | **82.05** | **86.30** | **81.84** |
| | PromptSRC | Vanilla | 85.43 | **80.28** | 82.77 | 74.86 | 14.06 | 14.74 | 14.39 | 11.95 | 63.56 | 55.60 | 59.31 | 55.41 |
| | | +ProTeCt | 85.36 | 78.72 | 81.91 | 73.82 | 64.76 | 40.02 | 49.47 | 42.36 | 91.19 | 79.38 | 84.88 | 80.53 |
| | | +Ours | **85.58** | **80.28** | **82.85** | **74.89** | **67.66** | **40.12** | **50.37** | **42.87** | **91.74** | **79.82** | **85.37** | **81.72** |

Table 3: Ablation results on Cifar100, SUN, and Rare Species using MaPLe under different $k$-shot settings. Additional ablation results and analyses are provided in Appendix C.

| K-Shot | Variant | Cifar100 | | | SUN | | | Rare Species | | |
|---|---|---|---|---|---|---|---|---|---|---|
| | | LA | HCA | MTA | LA | HCA | MTA | LA | HCA | MTA |
| 1 | +ProTeCt | 69.33 | 48.10 | 83.36 | 64.29 | 50.45 | 76.73 | 39.92 | 13.22 | 70.04 |
| | +Ours-Euc | 69.79 | 49.77 | 84.54 | 67.19 | 56.25 | 79.80 | 46.56 | 20.28 | 74.25 |
| | +Ours-HypV1 | 69.80 | 51.86 | 85.17 | 66.78 | 57.56 | 80.22 | 45.51 | 20.86 | 76.62 |
| | +Ours-HypV2 | 70.98 | 51.67 | 85.23 | 67.17 | 57.87 | 80.44 | 45.81 | 20.82 | 76.54 |
| | +Ours | **71.37** | **53.19** | **85.29** | **67.57** | **57.92** | **80.55** | **46.77** | **20.94** | **76.83** |
| 16 | +ProTeCt | 75.34 | 61.15 | 88.04 | 72.17 | 59.71 | 82.27 | 48.14 | 24.82 | 78.79 |
| | +Ours-Euc | 76.99 | 68.01 | 90.55 | 74.07 | 66.81 | 85.36 | 68.96 | 51.81 | 87.15 |
| | +Ours-HypV1 | 77.62 | 69.05 | 90.82 | 75.10 | 68.26 | 85.99 | 67.41 | 52.85 | 87.18 |
| | +Ours-HypV2 | 77.69 | 69.33 | 90.71 | 75.19 | 68.65 | 85.92 | 69.67 | 52.73 | 87.01 |
| | +Ours | **77.92** | **69.38** | **90.89** | **75.47** | **68.67** | **86.02** | **69.96** | **53.65** | **87.27** |

modalities share a common learnable curvature; and (iii) **Ours-HypV2**, in which each modality is assigned an independently learnable curvature. We conduct ablation in the few-shot setting, with results summarized in Table 3.

**Effectiveness of Semantic-Aware Visual Feature Extraction Framework.** Ours-Euc consistently outperforms ProTeCt. This gain results from our framework's ability to extract coarse-to-fine visual features and construct symmetric feature trees, facilitating symmetric alignment and enhancing performance.

**Effectiveness of Alignment on Hyperbolic Manifolds.** The comparison of Ours-Euc with the hyperbolic variants (Ours-HypV1, Ours-HypV2, and Ours) demonstrates that incorporating our hyperbolic alignment constraint helps better preserve hierarchical relationships.

**Effectiveness of Intermediate Manifold Search.** The comparisons among the hyperbolic variants highlight the importance of intermediate manifold search. By aligning visual and textual features through an optimized intermediate manifold, our approach achieves more effective hierarchical alignment across modalities.

**Efficiency Analysis.** We further measure the computational overhead of our method and that of learning a single shared curvature. The time cost for the single-curvature method and our method is 74 s and 74.5 s per batch, and the memory cost is 10,400 MB and 10,400 MB, respectively. The results demonstrate that the additional time and memory overhead introduced by learning multiple curvatures is negligible. More details are provided in Appendix C.16.

## 5.5 VISUALIZATION

**Visualization of learned representations.** Figure 5 shows that our semantic-aware visual feature extraction framework produces more separable hierarchical representations across all taxonomic

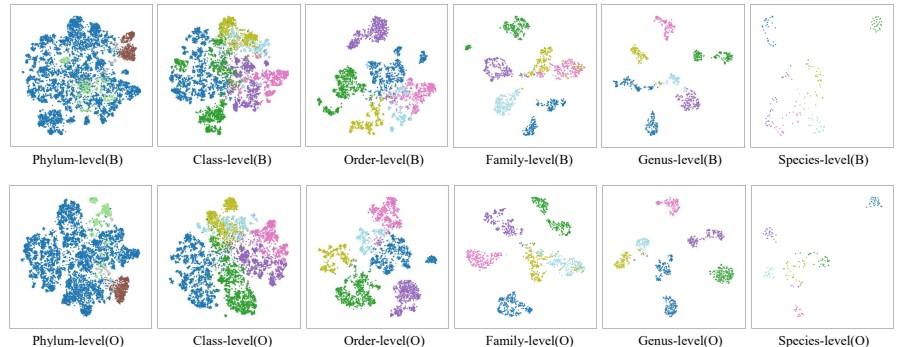

Figure 5: T-SNE visualization of learned image representations from the baseline ProTeCt (B) and our method (O), colored by taxonomic labels. Our method demonstrates improved feature separability across taxonomic categories. Additional quantitative results are provided in Appendix C.7

levels compared to ProTeCt. The clearer inter-class boundaries and compact intra-class distributions demonstrate the effectiveness of our method in extracting coarse-to-fine visual features that align with the hierarchical text structure, validating our tree-based alignment approach.

**Visualization of attention maps.** We use GradCAM (Selvaraju et al., 2017) to visualize the attention maps generated by our model to analyze its behavior across different taxonomic levels. As shown in Figure 6, when aligned with text prompts at different granularities, our model attends to distinct visual regions for the same image. For instance, when distinguishing at the class level (*e.g.*, mammal), the model focuses on features like fur, while at the genus level (*e.g.*, ailuropoda), it shifts attention to facial characteristics. This confirms that our semantic-aware feature extraction framework adaptively captures the most relevant visual cues for each taxonomic level, generating appropriate hierarchical features for alignment. See Appendices C.13 and C.14 for more details.

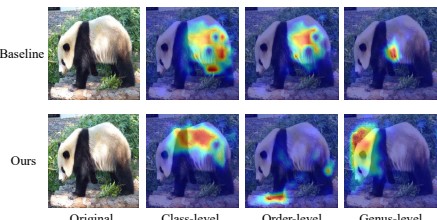

Figure 6: Visualization of attention maps of our model and the baseline ProTeCt across taxonomic levels.

## 6 CONCLUSION

In this work, we have presented an Alignment across Trees method to address asymmetric modality misalignment in vision–language models. The proposed method consists of a semantic-aware visual feature extraction framework and a heterogeneous manifold alignment algorithm. The framework leverages class tokens from intermediate Transformer layers and a text-guided cross-attention module to produce visual features with coarse-to-fine semantics. The tree-like visual and textual features are then embedded into distinct hyperbolic manifolds with various curvatures. The proposed heterogeneous manifold alignment algorithm constructs an intermediate manifold by formulating and minimizing the manifold distance, and aligns textual and visual features on this intermediate manifold. Extensive experiments on taxonomic open-set classification tasks demonstrate that our method extracts symmetric cross-modal features, captures and aligns their geometric structures on hyperbolic manifolds, leading to consistent improvements on various datasets and settings.

**Limitations and future work.** Considering the complexity of the underlying geometric structures, a single-curvature hyperbolic manifold is inherently limited in capturing such rich semantics. Moreover, our method does not explicitly account for the fact that hierarchical data in real-world scenarios often exhibit domain shifts. In future work, we plan to leverage more expressive mixed-curvature spaces to separately model visual and textual features, and we plan to develop a robust method that can handle cross-domain variations, improving the generalization in the open world.

ACKNOWLEDGEMENTS

This work was supported by the National Natural Science Foundation of China (NSFC; Grant No. 62406009), the Natural Science Foundation of Shenzhen (Grant No. JCYJ20230807142703006), the Shenzhen Science and Technology Program (Grant No. JCYJ20241202130548062), the Key Research Platforms and Projects of the Guangdong Provincial Department of Education (Grant No. 2023ZDZX1034), and the Opening Project of the State Key Laboratory of General Artificial Intelligence, BIGAI/Peking University, Beijing, China (Grant No. SKLAGI20250P07).

REPRODUCIBILITY STATEMENT.

To ensure the reproducibility of our work, we have made significant efforts to provide comprehensive implementation details and resources. We provide detailed training procedures and hyperparameters required to reproduce our experiments in Appendix B. Complete proofs for all theoretical contributions, including clear statements of assumptions and mathematical derivations, are provided in Appendix A. For the Rare Species dataset, which has not been previously used for TOS classification tasks, we provide data preprocessing steps and adaptation procedures in Appendix B.1. For our newly proposed base-to-base, base-to-novel, and base-to-whole evaluation settings, we provide detailed descriptions of the base/novel tree partitioning methodology and experimental configurations in Appendix B.5. We are committed to open-sourcing our code and trained models to facilitate future research. We believe these materials will enable researchers to reproduce our findings and further advance our work.

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

# A PROOFS

## A.1 PROOF OF THEOREM 1

We present the proof of theorem 1 as follows. We first restate the main setting for clarity. Let $\mathcal{L}^{c_1}$ and $\mathcal{L}^{c_3}$ be two manifolds. For clarity, $\mathcal{L}^{c_1}$ can be understood as the curvature of a manifold representing the distribution of textual and visual features, while $\mathcal{L}^{c_3}$ corresponds to the curvature of an intermediate manifold. The feature distributions on the two manifolds are given by

$$
\begin{aligned}
P_{c_1,\boldsymbol{u}_1} = \mathcal{N}_{\mathcal{L}}(\boldsymbol{x} \mid \boldsymbol{u}_1, \delta) = \frac{1}{Z(\delta)} \exp\left(-\frac{d_{c_1}^2(\boldsymbol{x}, \boldsymbol{u}_1)}{2\delta^2}\right), \\
P_{c_3,\boldsymbol{u}_3} = \mathcal{N}_{\mathcal{L}}(\boldsymbol{x} \mid \boldsymbol{u}_3, \delta) = \frac{1}{Z(\delta)} \exp\left(-\frac{d_{c_3}^2(\boldsymbol{x}, \boldsymbol{u}_3)}{2\delta^2}\right),
\end{aligned}
\tag{21}
$$

where $c_1$, $c_3$ represent the negative curvatures of the two hyperbolic manifolds, and $u_1$, $u_3$, $\delta$ are parameters of the Gaussian distributions. We assume the parameters of $\mathcal{L}^{c_1}$ and $P_{c_1,\boldsymbol{u}_1}$ (*i.e.*, $c_1$, $\boldsymbol{u_1}$, and $\delta$) are constant. The distance on hyperbolic manifolds can be computed as

$$
d_c^2(\boldsymbol{x}, \boldsymbol{u}) = \frac{1}{c} arccosh^2(-c\langle\boldsymbol{x}, \boldsymbol{u}\rangle_{\mathcal{L}}).
\tag{22}
$$

For simplicity, we approximate the distance function in Eq.(22) using a Taylor expansion around the point $y$.

$$
\begin{aligned}
&\frac{1}{c} \mathrm{arccosh}^2(-c\langle\boldsymbol{x}, \boldsymbol{u}\rangle_{\mathcal{L}}) \\
&\approx \frac{1}{c} \mathrm{arccosh}^2(y) + \frac{1}{c}\left[\mathrm{arccosh}^2\right]'(y)\left(-c\langle\boldsymbol{x}, \boldsymbol{u}\rangle_{\mathcal{L}} - y\right) \\
&= \frac{\mathrm{arccosh}^2(y) - \frac{2\mathrm{arccosh}(y)y}{\sqrt{y^2-1}}}{c} - \frac{2\mathrm{arccosh}(y)}{\sqrt{y^2-1}}\langle\boldsymbol{x}, \boldsymbol{u}\rangle_{\mathcal{L}}
\end{aligned}
\tag{23}
$$

We define the distance between $\mathcal{L}^{c_1}$ and $\mathcal{L}^{c_3}$ based on the Kullback-Leibler (KL) divergence:

$$
\begin{aligned}
\mathrm{KL}\left(P_{c_1,\boldsymbol{u}_1} \parallel P_{c_3,\boldsymbol{u}_3}\right) &= -\frac{1}{2\delta^2}\mathbb{E}_P\left[d_{c_1}^2(\boldsymbol{x}, \boldsymbol{u}_1)\right] + \frac{1}{2\delta^2}\mathbb{E}_P\left[d_{c_3}^2(\boldsymbol{x}, \boldsymbol{u}_3)\right] \\
&= R(c_1, \boldsymbol{u}_1, \delta) + \frac{1}{2\delta^2}d_{\mathcal{L}}(\mathcal{L}^{c_1}, \mathcal{L}^{c_3})
\end{aligned}
\tag{24}
$$

where $R(c_1, \boldsymbol{u}_1, \delta) = -\frac{1}{2\delta^2}\mathbb{E}_P\left[d_{c_1}^2(\boldsymbol{x}, \boldsymbol{u}_1)\right]$ is a constant and $d_{\mathcal{L}}(\mathcal{L}^{c_1}, \mathcal{L}^{c_3}) = \mathbb{E}_P\left[d_{c_3}^2(\boldsymbol{x}, \boldsymbol{u}_3)\right]$.

Thus, $d_{\mathcal{L}}(\mathcal{L}^{c_1}, \mathcal{L}^{c_3})$ is an affine transformation of the Kullback-Leibler (KL) divergence.

We observe that $d_{\mathcal{L}}(\mathcal{L}^{c_1}, \mathcal{L}^{c_3})$ is a function of $c_3$, and we approximate it using Taylor expansions (as shown in Eq.(23)):

$$d_{\mathcal{L}}(\mathcal{L}^{c_1}, \mathcal{L}^{c_3}) \approx d_{\mathcal{L}}(\mathcal{L}^{c_1}, \mathcal{L}^{c_3}; y_1)$$

$$= \mathbb{E}_P \left[ \frac{\mathrm{arccosh}^2(y_1) - \frac{2\,\mathrm{arccosh}(y_1)y_1}{\sqrt{y_1^2-1}}}{c_3} - \frac{2\,\mathrm{arccosh}(y_1)}{\sqrt{y_1^2-1}} \langle \boldsymbol{x}, \boldsymbol{u}_3 \rangle_{\mathcal{L}} \right]$$

$$= \frac{\mathrm{arccosh}^2(y_1) - \frac{2\,\mathrm{arccosh}(y_1)y_1}{\sqrt{y_1^2-1}}}{c_3} + \mathbb{E}_P \left[ -\frac{2\,\mathrm{arccosh}(y_1)}{\sqrt{y_1^2-1}} \langle \boldsymbol{x}, \boldsymbol{u}_3 \rangle_{\mathcal{L}} \right] \quad (25)$$

$$= \frac{\mathrm{arccosh}^2(y_1) - \frac{2\,\mathrm{arccosh}(y_1)y_1}{\sqrt{y_1^2-1}}}{c_3} + \frac{2\,\mathrm{arccosh}(y_1)}{\sqrt{y_1^2-1}} \left( -\langle \mathbb{E}_P[\boldsymbol{x}], \boldsymbol{u}_3 \rangle_{\mathcal{L}} \right)$$

$$= \frac{\mathrm{arccosh}^2(y_1) - \frac{2\,\mathrm{arccosh}(y_1)y_1}{\sqrt{y_1^2-1}}}{c_3} + \frac{2\,\mathrm{arccosh}(y_1)}{\sqrt{y_1^2-1}} \left( -\langle \boldsymbol{u}_1, \boldsymbol{u}_3 \rangle_{\mathcal{L}} \right)$$

Since $u_1$ and $u_3$ are mapped from midpoints in the tangent space (refer to Eq.(1) for the exponential map, where we use the tangent space at the origin), we can further expand the Lorentzian inner product:

$$-\langle \boldsymbol{u}_1, \boldsymbol{u}_3 \rangle_{\mathcal{L}} = \boldsymbol{u}_{1\text{time}} \boldsymbol{u}_{3\text{time}} - \langle \boldsymbol{u}_{1\text{space}}, \boldsymbol{u}_{3\text{space}} \rangle$$

$$= \sqrt{\frac{1}{c_1} + \|\boldsymbol{u}_{1\text{space}}\|^2} \sqrt{\frac{1}{c_3} + \|\boldsymbol{u}_{3\text{space}}\|^2} - \langle \boldsymbol{u}_{1\text{space}}, \boldsymbol{u}_{3\text{space}} \rangle$$

$$= \sqrt{\frac{1}{c_1} + \left( \frac{\sinh(\sqrt{c_1}\|\bar{\boldsymbol{v}}\|)}{\sqrt{c_1}\|\bar{\boldsymbol{v}}\|} \bar{\boldsymbol{v}} \right)^2} \sqrt{\frac{1}{c_3} + \left( \frac{\sinh(\sqrt{c_3}\|\bar{\boldsymbol{v}}\|)}{\sqrt{c_3}\|\bar{\boldsymbol{v}}\|} \bar{\boldsymbol{v}} \right)^2}$$

$$- \left\langle \frac{\sinh(\sqrt{c_1}\|\bar{\boldsymbol{v}}\|)}{\sqrt{c_1}\|\bar{\boldsymbol{v}}\|} \bar{\boldsymbol{v}}, \frac{\sinh(\sqrt{c_3}\|\bar{\boldsymbol{v}}\|)}{\sqrt{c_3}\|\bar{\boldsymbol{v}}\|} \bar{\boldsymbol{v}} \right\rangle$$

$$= \sqrt{\frac{\sinh^2(\sqrt{c_1}\|\bar{\boldsymbol{v}}\|) + 1}{c_1}} \sqrt{\frac{\sinh^2(\sqrt{c_3}\|\bar{\boldsymbol{v}}\|) + 1}{c_3}} - \frac{\sinh(\sqrt{c_1}\|\bar{\boldsymbol{v}}\|)\sinh(\sqrt{c_3}\|\bar{\boldsymbol{v}}\|)}{\sqrt{c_1 c_3}}$$

$$= \frac{\sqrt{(\sinh^2(\sqrt{c_1}\|\bar{\boldsymbol{v}}\|) + 1)(\sinh^2(\sqrt{c_3}\|\bar{\boldsymbol{v}}\|) + 1)} - \sinh(\sqrt{c_1}\|\bar{\boldsymbol{v}}\|)\sinh(\sqrt{c_3}\|\bar{\boldsymbol{v}}\|)}{\sqrt{c_1 c_3}}$$

$$= \frac{\cosh\sqrt{c_1}\bar{\boldsymbol{v}}\cosh\sqrt{c_3}\bar{\boldsymbol{v}} - \sinh\sqrt{c_1}\bar{\boldsymbol{v}}\sinh\sqrt{c_3}\bar{\boldsymbol{v}}}{\sqrt{c_1 c_3}}$$

$$= \frac{\cosh\left( (\sqrt{c_1} - \sqrt{c_3})\|\bar{\boldsymbol{v}}\| \right)}{\sqrt{c_1 c_3}}$$

$$(26)$$

where $\|\bar{\boldsymbol{v}}\|$ is the Euclidean norm of the midpoint of features in the tangent space, which depends on the choice of $c_1$ and $c_3$. We define $r = \|\bar{\boldsymbol{v}}\|$ as a constant value.

Thus, we have

$$d_{\mathcal{L}}(\mathcal{L}^{c_1}, \mathcal{L}^{c_3}) \approx d_{\mathcal{L}}(\mathcal{L}^{c_1}, \mathcal{L}^{c_3}; y_1)$$

$$= \frac{A(y_1)}{c_3} + B(y_1)\frac{\cosh\left( (\sqrt{c_1} - \sqrt{c_3})r \right)}{\sqrt{c_1 c_3}}, \quad (27)$$

where $A(y_1)$ and $B(y_1)$ are computed as

$$A(y_1) = \left( \mathrm{arccosh}^2(y_1) - \frac{2\,\mathrm{arccosh}(y_1)\,y_1}{\sqrt{y_1^2-1}} \right), B(y_1) = \frac{2\,\mathrm{arccosh}(y_1)}{\sqrt{y_1^2-1}}. \quad (28)$$

Notice that $d_{\mathcal{L}}(\mathcal{L}^{c_1}, \mathcal{L}^{c_3}; y_1)$ is a function of $c_3$, which we denote as

$$f(c_3; y_1, c_1, r) = d_{\mathcal{L}}(\mathcal{L}^{c_1}, \mathcal{L}^{c_3}; y_1) \quad (29)$$

Next, we focus on selecting the Taylor expansion point $y_1$. To ensure that $d_{\mathcal{L}}(\mathcal{L}^{c_1}, \mathcal{L}^{c_3}; y_1, c_1, r)$ is a good approximation of the distance function, we need to find a $y_1^{\star}$ that satisfies $f'(c_3; y_1^{\star}, c_1, r) = 0$, i.e.,

$$
\begin{aligned}
\frac{d}{dc_3} f(c_3; y_1^{\star}, c_1, r)\Big|_{c_3=c_1} &= -\frac{A(y_1^{\star})}{c_1^2} - \frac{B(y_1^{\star})}{2c_1^2} \\
&= -\frac{1}{c_1^2}\left(A(y_1^{\star}) + \frac{B(y_1^{\star})}{2}\right) \\
&= 0
\end{aligned}
\tag{30}
$$

This equation has a numerical solution $y_1^{\star} \approx 3.016$ and the corresponding $B(y_1^{\star}) > 0$

Thus, we have

$$
\begin{aligned}
&d_{\mathcal{L}}(\mathcal{L}^{c_1}, \mathcal{L}^{c_3}; y_1) \\
&= f(c_3; y_1^{\star}, c_1, r) \\
&= B(y_1^{\star})\left[\frac{-\sqrt{c_1} + 2\sqrt{c_3}cosh[(\sqrt{c_3} - \sqrt{c_1})r]}{2\sqrt{c_1}c_3}\right]
\end{aligned}
\tag{31}
$$

Let $D_{\mathcal{L}}(\mathcal{L}^{c_1}, \mathcal{L}^{c_3}) = \frac{-\sqrt{c_1} + 2\sqrt{c_3}cosh[(\sqrt{c_3} - \sqrt{c_1})r_1]}{2\sqrt{c_1}c_3}$, then we have

$$
\begin{aligned}
D_{\mathcal{L}}(\mathcal{L}^{c_1}, \mathcal{L}^{c_3}) &= d_{\mathcal{L}}(\mathcal{L}^{c_1}, \mathcal{L}^{c_3}; y_1)/B(y_1^{\star}) \\
&= \frac{\mathrm{KL}\left(P_{c_1, \boldsymbol{u}_1} \parallel P_{c_3, \boldsymbol{u}_3}\right) - R(c_1, \boldsymbol{u}_1, \delta)}{2\delta^2 B(y_1^{\star})},
\end{aligned}
\tag{32}
$$

which indicates that our $D_{\mathcal{L}}(\mathcal{L}^{c_1}, \mathcal{L}^{c_3})$ is an affine transformation of the Kullback-Leibler (KL) divergence and the coefficient $\frac{1}{2\delta^2 B(y_1^{\star})}$ is positive.

The proposed distance is derived from the KL divergence, which does not satisfy symmetry or the triangle inequality. Consequently, the proposed distance does not satisfy these properties either. In the future, we consider leveraging JS divergence to derive the formal distance metric between manifolds.

## A.2 PROOF OF PROPOSITION 1

We just need to prove that $f(c_3; y_1^{\star}, c_1, r)$ is monotonically decreasing on $(0, c_1]$ and monotonically increasing on $[c_1, \infty)$.

To analyze the monotonicity of $f(c_3; y_1^{\star}, c_1, r)$, we examine its second derivative:

$$
\frac{\partial^2}{\partial c_3^2} f = \frac{\sqrt{c_3}(3 + r^2 c_3) \cosh\left[r(\sqrt{c_3} - \sqrt{c_1})\right] - 3c_3 r \sinh\left[r(\sqrt{c_3} - \sqrt{c_1})\right] - 4\sqrt{c_1}}{4\sqrt{c_1}c_3^3}.
\tag{33}
$$

The denominator is always positive for $c_3 > 0$. Defining the numerator as:

$$
N(c_3) = \sqrt{c_3}(3 + r^2 c_3) \cosh\left[r(\sqrt{c_3} - \sqrt{c_1})\right] - 3c_3 r \sinh\left[r(\sqrt{c_3} - \sqrt{c_1})\right] - 4\sqrt{c_1},
\tag{34}
$$

we require $N(c_3) > 0$ for all $c_3 \geq c_{\min} > 0$, where $c_{\min}$ is a positive lower bound for curvatures $c_1, c_2, c_3$.

POSITIVITY ANALYSIS OF $N(c_3)$

Let $d = \sqrt{c_3} - \sqrt{c_1}$ and $L = \sqrt{c_1} - \sqrt{c_{\min}} > 0$. For large $r$:

**Case 1:** $c_3 \geq c_1$ $(d \geq 0)$
The dominant term is $\frac{1}{2}\left[\sqrt{c_3}(3 + r^2 c_3) - 3c_3 r\right] e^{rd}$. Its coefficient is positive since:

$$
\sqrt{c_3}(3 + r^2 c_3) - 3c_3 r = 3\sqrt{c_3} + r^2 c_3^{3/2} - 3c_3 r > 0 \quad \forall r > 0, c_3 > 0.
\tag{35}
$$

The minimum occurs at $c_3 = c_1$:

$$
N(c_1) = \sqrt{c_1}(r^2 c_1 - 1) \geq 0 \quad \text{when} \quad r \geq \frac{1}{\sqrt{c_1}}.
\tag{36}
$$

**Case 2:** $c_{\min} \leq c_3 < c_1$ $(d < 0)$

The dominant term is $\frac{1}{2}\left[\sqrt{c_3}(3 + r^2 c_3) + 3c_3 r\right] e^{-rd}$ with $-d \geq L > 0$. Its coefficient is always positive. At $c_3 = c_{\min}$:

$$N(c_{\min}) = \sqrt{c_{\min}}(3 + r^2 c_{\min}) \cosh(rL) + 3c_{\min} r \sinh(rL) - 4\sqrt{c_1}. \tag{37}$$

For $rL \geq 2$, we have $\cosh(rL) \geq e^{rL}/3$ and $\sinh(rL) \geq e^{rL}/3$, leading to:

$$N(c_{\min}) \geq \frac{1}{3}\left[\sqrt{c_{\min}}(3 + r^2 c_{\min}) + 3c_{\min} r\right] e^{rL} - 4\sqrt{c_1} \tag{38}$$

$$\geq \frac{1}{3} c_{\min}^{3/2} r^2 e^{rL} - 4\sqrt{c_1}. \tag{39}$$

This is positive when:

$$\frac{1}{3} c_{\min}^{3/2} r^2 e^{rL} > 4\sqrt{c_1} \implies rL > \ln\left(\frac{12\sqrt{c_1}}{c_{\min}^{3/2} L^2}\right). \tag{40}$$

SUFFICIENT CONDITION FOR $r$

Define $L = \sqrt{c_1} - \sqrt{c_{\min}} > 0$. For $r > r_{\min}$ with:

$$r_{\min} = \max\left\{\frac{1}{\sqrt{c_1}}, \frac{2}{L}, \frac{1}{L}\ln\left(\frac{12\sqrt{c_1}}{c_{\min}^{3/2} L^2}\right)\right\}, \tag{41}$$

we have $N(c_3) > 0$ for all $c_3 \geq c_{\min}$. Thus:

$$\frac{\partial^2}{\partial c_3^2} f(c_3; y_1^\star, c_1, r) > 0 \quad \forall c_3 \geq c_{\min}. \tag{42}$$

MONOTONICITY CONCLUSION

The function $f$ is strictly convex for $c_3 \geq c_{\min}$. Combined with the first derivative analysis:

- At $c_3 = c_1$, $\frac{\partial f}{\partial c_3} = 0$
- For $c_3 < c_1$ $(c_3 \geq c_{\min})$, $\frac{\partial f}{\partial c_3} < 0$
- For $c_3 > c_1$, $\frac{\partial f}{\partial c_3} > 0$

Thus $f(c_3; y_1^\star, c_1, r)$ is monotonically decreasing on $[c_{\min}, c_1]$ and monotonically increasing on $[c_1, \infty)$. Thus, Proposition 1 holds.

### A.3 PROOF OF PROPOSITION 2

When $c_1 = c_2$, proposition 2 holds trivially. For $c_1 \neq c_2$, without loss of generality assume $c_1 < c_2$. We prove the existence and uniqueness of the minimizer $c_3^* \in [c_1, c_2]$ by analyzing the first derivative of $L_D(c_3)$:

$$\frac{dJ_c}{dc_3} = \frac{C}{2\sqrt{c_2}c_3^2} + \frac{D}{2\sqrt{c_1}c_3^2}, \tag{43}$$

where $C = \sqrt{c_2} - \sqrt{c_3}\cosh[(\sqrt{c_3} - \sqrt{c_2})r] + c_3 r \sinh[(\sqrt{c_3} - \sqrt{c_2})r]$ and $D = \sqrt{c_1} - \sqrt{c_3}\cosh[(\sqrt{c_3} - \sqrt{c_1})r] + c_3 r \sinh[(\sqrt{c_3} - \sqrt{c_1})r]$.

Define $M = \sqrt{c_2} - \sqrt{c_1} > 0$. At the endpoint $c_3 = c_1$:

$$\left.\frac{dL_D}{dc_3}\right|_{c_3=c_1} = \frac{\sqrt{c_2} - \sqrt{c_1}\cosh(Mr) - c_1 r \sinh(Mr)}{2\sqrt{c_2}c_1^2}.$$

For $r > \max\left\{\frac{3}{\sqrt{c_2}}, \frac{4}{M}\right\}$, the numerator is strictly negative. At $c_3 = c_2$:

$$\left.\frac{dL_D}{dc_3}\right|_{c_3=c_2} = \frac{\sqrt{c_1} - \sqrt{c_2}\cosh(Mr) + c_2 r \sinh(Mr)}{2\sqrt{c_1}c_2^2},$$

which is strictly positive when $r > \frac{3}{\sqrt{c_2}}$. By continuity of the derivative and the intermediate value theorem, there exists $c_3^\star \in (c_1, c_2)$ where the derivative vanishes.

To establish uniqueness, we extend the curvature bound from appendix A.2. Define the enhanced radius threshold:

$$r_{\min}^* = \max \left\{ r_{\min}, \ \frac{1}{\sqrt{c_2}}, \ \frac{2}{\sqrt{c_2} - \sqrt{c_{\min}}}, \ \frac{1}{M_{\min}} \ln \left( \frac{12\sqrt{c_2}}{c_{\min}^{3/2} M_{\min}^2} \right) \right\}, \tag{44}$$

where $M_{\min} = \sqrt{c_2} - \sqrt{c_{\min}}$. For $r \geq \max \left\{ r_{\min}^*, \frac{4}{M}, \frac{3}{\sqrt{c_2}} \right\}$, the second derivative $\frac{\mathrm{d}^2 L_p}{\mathrm{d} c_3^2} > 0$ throughout $[c_1, c_2]$ (proof methodology identical to appendix A.2). This strict convexity guarantees a unique minimizer $c_3^* \in [c_1, c_2]$.

Thus proposition 2 holds for all cases.

## B    EXPERIMENTAL SETTINGS

### B.1    DATASETS

Our experiments are conducted on four datasets: Cifar100 (Krizhevsky et al., 2009), SUN (Xiao et al., 2010), ImageNet (Deng et al., 2009), and Rare Species (Stevens et al., 2024). Cifar100 is a dataset containing 100 classes of images, each with 600 samples, designed for fine-grained classification tasks. SUN is a scene recognition dataset that includes 397 scene categories, representing a wide variety of indoor and outdoor environments. ImageNet is a large-scale dataset with millions of labeled images across 1,000 categories, primarily used for large-scale image classification tasks. Rare Species, on the other hand, focuses on rare species classification and provides hierarchical annotations spanning multiple taxonomic levels, including kingdom, phylum, class, order, family, genus, and species.

It is worth noting that Cifar100, SUN, and ImageNet do not natively include hierarchical labels. However, these datasets were extended into hierarchical versions by Wu et al. (2024) using a generic public taxonomy (*e.g.*, WordNet (Fellbaum, 1998)) or a specialized taxonomy related to the application, such as scientific taxonomies. In contrast, Rare Species offers a more rigorous hierarchical taxonomy. Each sample is annotated at multiple levels, from kingdom to species, with well-defined and consistent hierarchical relationships. Unlike the other datasets, where hierarchical labels are inferred or constructed from external sources, Rare Species ensures that every leaf node is at the same depth, and each sample has a corresponding label at every level. This consistency makes Rare Species particularly suited for TOS classification.

To prepare the Rare Species dataset for TOS classification, we construct a hierarchical tree structure based on the taxonomic annotations. The Rare Species dataset provides annotations at seven taxonomic levels for each image: kingdom, phylum, class, order, family, genus, and species. A key challenge in constructing this hierarchy is that identical species names may refer to different organisms across different taxonomic lineages. Specifically, there are 28 out of 400 species names that refer to multiple organisms. To address this ambiguity, we create unique identifiers by concatenating all seven taxonomic levels, ensuring each leaf node represents a distinct biological entity. This also aligns with the BIOCLIP's text style during pretraining. Additionally, since all samples in the Rare Species dataset belong to the animal kingdom (Animalia), we skip the grouping process at the kingdom level during tree construction. The complete preprocessing procedure is detailed in Algorithm 1. We also consider an alternative approach where we use the common name accompanied by the species name to handle ambiguity at the species level, as discussed in Appendix C.11.

### B.2    METRICS

We evaluate TOS classification performance using three metrics proposed by Wu et al. (2024): Leaf Accuracy (LA), Hierarchical Consistent Accuracy (HCA), and Mean Treecut Accuracy (MTA). We include here for completeness the definitions introduced in Wu et al. (2024).

**Problem Formulation.**    A class taxonomy $\mathcal{Y}_{\text{tax}}$ organizes classes into a tree where classes of similar semantics are recursively assembled into superclasses at each graph node. For an image $x$, a

---

**Algorithm 1** Hierarchical Tree Construction for Rare Species Dataset

---

**Require:** Dataset $\mathcal{D}$ with taxonomic annotations for each image
**Ensure:** Hierarchical tree structure $\mathcal{T}$
 1: Initialize empty tree $\mathcal{T}$ and empty dictionary nodes
 2: **Step 1: Create unique identifiers for leaf nodes**
 3: **for** each image $i$ in $\mathcal{D}$ **do**
 4:     Extract taxonomic labels: $\{k_i, p_i, c_i, o_i, f_i, g_i, s_i\}$
 5:     Create unique identifier: $\text{uid}_i \leftarrow \text{concat}(k_i, p_i, c_i, o_i, f_i, g_i, s_i)$
 6:     Create leaf node with $\text{uid}_i$ and add to $\text{nodes}[\text{uid}_i]$
 7: **end for**
 8: **Step 2: Recursively build internal nodes**
 9: **for** level $\ell$ from 6 to 1 **do**
10:     {From genus to phylum (skip kingdom)}
11:     Group nodes by taxonomic prefix at level $\ell$
12:     **for** each unique prefix $\text{prefix}_\ell$ at level $\ell$ **do**
13:         children $\leftarrow$ all nodes with matching $\text{prefix}_\ell$
14:         Create parent node with identifier $\text{prefix}_\ell$
15:         Connect parent to all nodes in children
16:         Add parent node to $\text{nodes}[\text{prefix}_\ell]$
17:     **end for**
18: **end for**
19: **Return** Tree structure $\mathcal{T}$ with hierarchical taxonomy

---

classifier predicts a label given the label set $\mathcal{Y}$ and model parameters $\boldsymbol{\theta}$:

$$\hat{y}(\boldsymbol{x}; \mathcal{Y}, \boldsymbol{\theta}) = \arg\max_{t_y \in \mathcal{Y}} p(t_y \mid \boldsymbol{x}; \mathcal{Y}, \boldsymbol{\theta}). \tag{45}$$

**Leaf Accuracy (LA).**   It is defined as

$$\text{LA} = \frac{1}{N} \sum_{i=1}^{N} \mathbb{1}[\hat{y}(\boldsymbol{x}_i; \mathcal{Y}_{\text{leaf}}) = t_{y_i}], \tag{46}$$

where $t_{y_i}$ denotes the leaf node corresponding to groundtruth label $y_i$ of the i-th image $\boldsymbol{x}_i$ It measures the classification accuracy at the leaves of the taxonomic tree. This enables comparison of hierarchical classifiers to standard, or flat, classifiers which only consider the leaf classes.

**Hierarchical Consistent Accuracy (HCA).**   It is defined as

$$\text{HCA} = \frac{1}{N} \sum_{i=1}^{N} \left( \mathbb{1}[\hat{y}(\boldsymbol{x}_i; \mathcal{Y}_{\text{leaf}}) = t_{y_i}] \prod_{n \in \mathcal{A}(t_{y_i})} \mathbb{1}[\hat{y}(\boldsymbol{x}_i; \mathcal{Y}_n) \in \mathcal{A}(t_{y_i}) \cup \{t_{y_i}\}] \right), \tag{47}$$

where $\mathcal{A}(n)$ denotes all the ancestors of node $n$. While LA considers successful any correct classification at the leaf level of the tree, the HCA is stricter. It declares a success only when all the ancestors of the leaf node are correctly classified. In other words, each sample needs to be classified correctly at each tree level to be viewed as correctly classified under the HCA. LA is an upper bound for the HCA.

**Mean Treecut Accuracy (MTA).**   It estimates the expected accuracy under the TOS classification settings. It computes the average accuracy over a set of treecuts $\mathcal{T}_c \in \Omega$:

$$\text{MTA} = \frac{1}{|\Omega|} \sum_{\mathcal{T}_c \in \Omega} \frac{1}{N} \sum_{i=1}^{N} \mathbb{1}[\hat{y}(\boldsymbol{x}_i; \mathcal{Y}_{\mathcal{T}_c}) = t_{y_i}]. \tag{48}$$

Note that the number of treecuts of a tree is very large. It is impossible to evaluate on all treecuts of a given tree. Following Wu et al. (2024), we randomly sample $|\Omega| = 25$ treecuts from $\mathcal{T}$ in all experiments. The treecuts are generated once and used in the evaluation of all methods, thus ensuring fairness.

## B.3 TRAINING PROCEDURES

---

**Algorithm 2** Training Process of Heterogeneous Manifold Alignment (One Iteration)

---

**Require:** soft prompts $\theta$, learnable curvatures $c_1, c_2$, learning rate $\eta$, weight of entailment loss $\alpha$
**Ensure:** Updated parameters $\theta, c_1, c_2$
1: **Step 1: Construct intermediate manifold**
2: $c_3^\star \leftarrow \text{GoldenSectionSearch}(L_D(\cdot; c_1, c_2))$ {Eq.(10)}
3: **Step 2: Compute implicit gradients**
4: $\frac{\partial c_3^\star}{\partial c_1}, \frac{\partial c_3^\star}{\partial c_2} \leftarrow \text{ImplicitGradient}(L_D(\cdot; c_1, c_2), c_3^\star)$ {Eq.(20)}
5: **Step 3: Extract and map features**
6: $\{v_i\}_{i=1}^{H}, \{t_i\}_{i=1}^{H} \leftarrow \text{ExtractFeatures}()$ {Sec. 4.1}
7: **for** $i = 1$ to $H$ **do**
8: $\quad t_i^{c_1} \leftarrow \text{expm}_0^{c_1}(t_i), v_i^{c_2} \leftarrow \text{expm}_0^{c_2}(v_i)$
9: $\quad t_i^{c_3} \leftarrow \text{expm}_0^{c_3}(t_i), v_i^{c_3} \leftarrow \text{expm}_0^{c_3}(v_i)$
10: **end for**
11: **Step 4: Compute losses**
12: $J(\theta, c_1, c_2) = J_{\text{pro}}(T_e, V_e) + \alpha \left( J_{\text{Tent}}(T^{c_1}) + J_{\text{Vent}}(V^{c_2}) + J_{\text{ent}}(V^{c_3^*}, T^{c_3^*}) \right)$ {Eq.(17)}
13: **Step 5: Update parameters**
14: $\theta \leftarrow \theta - lr \cdot \nabla_\theta J$
15: $\frac{dJ}{dc_1} \leftarrow \frac{\partial J}{\partial c_1} + \frac{\partial J}{\partial c_3^*} \frac{\partial c_3^*}{\partial c_1}$ {Total derivative}
16: $\frac{dJ}{dc_2} \leftarrow \frac{\partial J}{\partial c_2} + \frac{\partial J}{\partial c_3^*} \frac{\partial c_3^*}{\partial c_2}$ {Total derivative}
17: $c_1 \leftarrow c_1 - lr \cdot \frac{dJ}{dc_1}$
18: $c_2 \leftarrow c_2 - lr \cdot \frac{dJ}{dc_2}$
19: **return** $\theta, c_1, c_2$

---

## B.4 TOS CLASSIFICATION

For TOS classification experiments on Cifar100, SUN, and ImageNet, our training settings are largely aligned with Wu et al. (2024) to ensure a fair comparison with baseline methods. For Rare Species, we use consistent settings across all baseline reproductions and our method experiments.

**Training epochs.** On Cifar100 and SUN, both MaPLe and PromptSRC are trained for 200 epochs with batch sizes of 128 and 32, respectively. On ImageNet and Rare Species, both models are trained for 10 epochs with batch sizes of 2 and 8, respectively.

**Model initialization.** We use a pretrained ViT-B/16 CLIP model for initialization. For Cifar100, SUN, and ImageNet, we use OpenAI's CLIP checkpoint. For Rare Species, we use BioCLIP (Stevens et al., 2024) checkpoint as the OpenAI pretrained CLIP performs poorly on this dataset.

**Choice of intermediate layers.** Across all experiments, we choose the $4^{\text{th}}$ layer, $7^{\text{th}}$ layer (counting from 0) as the intermediate layers for our Semantic Aware Feature Extraction framework introduced in Section 4.1.

**Optimization.** All experiments use the SGD optimizer with a cosine learning rate scheduler. For MaPLe experiments on Cifar100, SUN, and ImageNet, we adopt a learning rate of 0.02 to ensure a fair comparison with ProTeCt (Wu et al., 2024). For other experiments, the learning rate is chosen from $\{0.01, 0.02, 0.03\}$ depending on the prompt learning method and the dataset.

**Other hyperparameters.** Across all experiments, we use a fixed weight $\alpha = 0.5$ for the entailment loss. The initial values of curvatures for visual and textual manifolds are chosen from $\{0.5, 0.25, 0.05, 0.025\}$ depending on the dataset.

**Data preprocessing.** For Cifar100, SUN, and ImageNet, we use the preprocessed data provided by Wu et al. (2024). For Rare Species, the preprocessing procedure is depicted in Appendix B.1.

For all experiments, we report LA, HCA, and MTA, averaged over three independent runs.

### B.5 BASE-TO-NOVEL GENERALIZATION OF TOS CLASSIFICATION

To construct base and novel tree pairs, we first partition the leaf nodes of the semantic tree equally into two disjoint subsets: **base leaves** $\mathcal{Y}_B$ and **novel leaves** $\mathcal{Y}_N$. For each subset $\mathcal{Y}_S$ (where $S \in \{B, N\}$), we define the corresponding **base/novel tree** as the subgraph formed by the union of all root-to-leaf paths for every leaf $v \in \mathcal{Y}_S$. Detailed preprocessing procedure is illustrated in Algorithm 3 This construction yields a rooted subtree satisfying:

(a) The leaf set equals $\mathcal{Y}_S$ exactly.

(b) Internal nodes comprise all ancestors of $\mathcal{Y}_S$

(c) Edges preserve the original ancestor-descendant. relationships.

---

**Algorithm 3** Construction of Base and Novel Trees

---

**Require:** Semantic tree $\mathcal{T} = (\mathcal{V}, \mathcal{E})$ with root $r$ and leaf set $\mathcal{Y}$
**Ensure:** Base tree $\mathcal{T}_B$ and novel tree $\mathcal{T}_N$
  1: **Step 1: Partition leaf nodes**
  2: Partition $\mathcal{Y}$ into two equal disjoint subsets: $\mathcal{Y}_B$ and $\mathcal{Y}_N$
  3:    where $|\mathcal{Y}_B| = |\mathcal{Y}_N| = |\mathcal{Y}|/2$ and $\mathcal{Y}_B \cap \mathcal{Y}_N = \emptyset$
  4: **Step 2: Construct base tree $\mathcal{T}_B$**
  5: Initialize $\mathcal{V}_B \leftarrow \emptyset, \mathcal{E}_B \leftarrow \emptyset$
  6: **for** each leaf $v \in \mathcal{Y}_B$ **do**
  7:    Trace path $P_v$ from root $r$ to leaf $v$
  8:    $\mathcal{V}_B \leftarrow \mathcal{V}_B \cup \{$all nodes in $P_v\}$
  9:    $\mathcal{E}_B \leftarrow \mathcal{E}_B \cup \{$all edges in $P_v\}$
 10: **end for**
 11: $\mathcal{T}_B \leftarrow (\mathcal{V}_B, \mathcal{E}_B)$
 12: **Step 3: Construct novel tree $\mathcal{T}_N$**
 13: Initialize $\mathcal{V}_N \leftarrow \emptyset, \mathcal{E}_N \leftarrow \emptyset$
 14: **for** each leaf $v \in \mathcal{Y}_N$ **do**
 15:    Trace path $P_v$ from root $r$ to leaf $v$
 16:    $\mathcal{V}_N \leftarrow \mathcal{V}_N \cup \{$all nodes in $P_v\}$
 17:    $\mathcal{E}_N \leftarrow \mathcal{E}_N \cup \{$all edges in $P_v\}$
 18: **end for**
 19: $\mathcal{T}_N \leftarrow (\mathcal{V}_N, \mathcal{E}_N)$
 20:
 21: **return** $\mathcal{T}_B, \mathcal{T}_N$

---

For base-to-novel generalization experiments, we adopt the same settings as standard TOS classification experiments but modify only the hyperparameters. Specifically, we train all methods for 10 epochs across all datasets to maintain generalizability, as we observe that training for more epochs leads to a dramatic decrease in performance on the novel trees. For each of the metrics (*i.e.*, LA, HCA, MTA), we report results on the base tree, novel tree, and their harmonic mean, defined as:

$$Metric_{\text{harmony}} = \frac{2 \cdot Metric_{\text{base}} \cdot Metric_{\text{novel}}}{Metric_{\text{base}} + Metric_{\text{novel}}}, \tag{49}$$

where $Metric \in \{\text{LA}, \text{HCA}, \text{MTA}\}$.

## C   MORE EXPERIMENTAL RESULTS

### C.1   COMPARISONS WITH HYPERBOLIC ALIGNMENT METHODS

We compare our method with hyperbolic alignment methods, MERU (Desai et al., 2023) and Hy-CoCLIP (Pal et al., 2025). Results are presented in Table 4. Compared to the state-of-the-art HyCo-CLIP method, our method achieves up to a $25.89\%$ improvement on LA, a $41.17\%$ improvement on HCA, and a $50.79\%$ improvement on MTA. The results demonstrate the limited ability of hyperbolic methods in aligning modalities with hierarchical semantic structures, highlighting the necessity of our method.

Table 4: TOS classification results across three datasets compared with hyperbolic alignment methods.

| Method | Cifar100 | | | SUN | | | ImageNet | | |
|---|---|---|---|---|---|---|---|---|---|
| | LA | HCA | MTA | LA | HCA | MTA | LA | HCA | MTA |
| MERU | 48.57 | 8.42 | 37.72 | 53.67 | 20.60 | 45.88 | 39.35 | 1.17 | 40.03 |
| HYCOCLIP | 60.21 | 9.31 | 42.96 | 58.40 | 26.52 | 48.98 | 45.74 | 0.98 | 38.82 |
| Ours | **78.9** | **68.47** | **91.12** | **76.54** | **69.18** | **86.20** | **71.63** | **42.15** | **89.61** |

Table 5: Base-to-base/base-to-novel/base-to-whole generalization results across four datasets.

| Dataset | Base Method | Variant | LA | | | | HCA | | | | MTA | | | |
|---|---|---|---|---|---|---|---|---|---|---|---|---|---|---|
| | | | Base | Novel | HM | Whole | Base | Novel | HM | Whole | Base | Novel | HM | Whole |
| SUN | MaPLe | Vanilla | 80.77 | 76.85 | 78.76 | 69.45 | 38.51 | 37.62 | 38.06 | 33.31 | 65.23 | 61.61 | 63.37 | 55.82 |
| | | +ProTeCt | 81.77 | 76.67 | 79.14 | 69.80 | 64.27 | 55.43 | 59.52 | 53.74 | 85.30 | 81.10 | 83.15 | 76.37 |
| | | +Ours | **82.79** | **77.11** | **79.85** | **69.82** | **73.38** | **56.23** | **63.67** | **57.09** | **88.85** | **81.24** | **84.87** | **78.61** |
| | PromptSRC | Vanilla | 82.30 | 78.68 | 80.58 | 71.50 | 51.77 | 48.25 | 49.94 | 47.05 | 68.89 | 65.67 | 67.24 | 58.93 |
| | | +ProTeCt | 82.36 | 78.40 | 80.33 | 71.94 | 66.86 | 58.80 | 62.57 | 57.06 | 86.67 | 82.76 | 84.67 | 79.12 |
| | | +Ours | **83.40** | **78.72** | **80.99** | **72.04** | **73.11** | **59.10** | **65.36** | **58.42** | **89.02** | **82.86** | **85.83** | **79.74** |
| Cifar100 | MaPLe | Vanilla | 82.60 | 75.80 | 79.05 | 71.45 | 9.94 | 6.94 | 8.17 | 7.00 | 61.46 | 50.66 | 55.54 | 54.14 |
| | | +ProTeCt | 82.66 | 74.56 | 78.40 | 70.03 | 61.84 | 35.86 | 45.40 | 39.37 | 89.65 | 77.15 | 82.93 | 77.34 |
| | | +Ours | **82.76** | **75.94** | **79.20** | **72.52** | **66.42** | **41.52** | **51.10** | **45.79** | **91.01** | **82.05** | **86.30** | **81.84** |
| | PromptSRC | Vanilla | 85.43 | **80.28** | 82.77 | 74.86 | 14.06 | 14.74 | 14.39 | 11.95 | 63.56 | 55.60 | 59.31 | 55.41 |
| | | +ProTeCt | 85.36 | 78.72 | 81.91 | 73.82 | 64.76 | 40.02 | 49.47 | 42.36 | 91.19 | 79.38 | 84.88 | 80.53 |
| | | +Ours | **85.58** | **80.28** | **82.85** | **74.89** | **67.66** | **40.12** | **50.37** | **42.87** | **91.74** | **79.82** | **85.37** | **81.72** |
| ImageNet | MaPLe | Vanilla | 78.59 | 75.78 | 77.16 | 67.50 | 1.84 | 1.96 | 1.90 | 1.64 | 48.57 | 45.85 | 47.17 | 45.10 |
| | | +ProTeCt | 78.23 | 75.42 | 76.80 | 67.48 | 33.92 | 29.28 | 31.43 | 27.01 | 90.86 | 88.02 | 89.42 | 86.61 |
| | | +Ours | **78.74** | **75.88** | **77.28** | **67.53** | **50.81** | **31.70** | **39.04** | **33.60** | **92.09** | **88.11** | **90.06** | **86.73** |
| | PromptSRC | Vanilla | 79.67 | 77.12 | 78.37 | 68.70 | 5.26 | 3.53 | 4.22 | 3.78 | 53.57 | 51.71 | 52.62 | 52.33 |
| | | +ProTeCt | 79.52 | 77.02 | 78.25 | 68.67 | 40.01 | 29.57 | 34.01 | 29.90 | 91.10 | 87.90 | 89.47 | 86.81 |
| | | +Ours | **79.80** | **77.15** | **78.45** | **68.74** | **49.59** | **30.32** | **37.63** | **33.31** | **92.03** | **88.08** | **90.01** | **86.95** |
| Rare Species | MaPLe | Vanilla | 52.95 | 47.83 | 50.26 | 43.93 | 9.25 | 7.78 | 8.45 | 5.72 | 50.45 | 52.94 | 51.67 | 45.52 |
| | | +ProTeCt | 52.61 | 47.32 | 49.82 | 44.18 | 29.94 | 16.14 | 20.97 | 17.15 | 79.72 | 72.32 | 75.84 | 72.81 |
| | | +Ours | **68.54** | **48.01** | **56.47** | **53.02** | **55.00** | **17.22** | **26.23** | **29.79** | **88.04** | **74.18** | **80.52** | **78.62** |
| | PromptSRC | Vanilla | 57.28 | 52.40 | 54.73 | 50.15 | 17.07 | 11.09 | 13.45 | 10.43 | 63.04 | 57.58 | 60.19 | 55.90 |
| | | +ProTeCt | 57.51 | 53.30 | 55.33 | 50.44 | 40.69 | 21.01 | 27.71 | 24.57 | 82.75 | 77.19 | 79.87 | 76.98 |
| | | +Ours | **64.93** | **53.72** | **58.80** | **53.32** | **51.89** | **21.11** | **30.01** | **29.29** | **87.70** | **77.27** | **82.16** | **78.79** |

## C.2 Complete Results of Base-to-Base/Base-to-Novel/Base-to-Whole Generalization

Table 5 presents the complete results of base-to-base/base-to-novel/base-to-whole generalization results across multiple datasets. between our method and existing methods. Our method consistently outperforms all baselines across all metrics and settings, demonstrating strong generalization to novel classes.

## C.3 More ablation results on TOS classification.

Table 6 summarizes our ablation results on Cifar100, SUN, and Rare Species using MaPLe and PromptSRC under 1-shot and 16-shot settings. Ours-Euc consistently outperforms ProTeCt, demonstrating the effectiveness of our semantic-aware visual feature extraction framework. Our method consistently outperforms two hyperbolic variants as well as Ours-Euc, demonstrating the effectiveness of our heterogeneous manifold alignment algorithm.

We also conduct ablation experiments under base-to-base/base-to-novel/base-to-whole settings on the SUN dataset. As shown in Table 7, we observe similar performance patterns, where our method consistently achieves superior results across different evaluation protocols. These results validate the generalization capability of our proposed approach.

Table 6: Ablation results on Cifar100, SUN, and Rare Species under different $k$-shot settings (Extended).

| K-Shot | Base Method | Variant | Cifar100 | | | SUN | | | Rare Species | | |
|---|---|---|---|---|---|---|---|---|---|---|---|
| | | | LA | HCA | MTA | LA | HCA | MTA | LA | HCA | MTA |
| 1 | MaPLe | +ProTeCt | 69.33 | 48.10 | 83.36 | 64.29 | 50.45 | 76.73 | 39.92 | 13.22 | 70.04 |
| | | +Ours-Euc | 69.79 | 49.77 | 84.54 | 67.19 | 56.25 | 79.80 | 46.56 | 20.28 | 74.25 |
| | | +Ours-HypV1 | 69.80 | 51.86 | 85.17 | 66.78 | 57.56 | 80.22 | 45.51 | 20.86 | 76.62 |
| | | +Ours-HypV2 | 70.98 | 51.67 | 85.23 | 67.17 | 57.87 | 80.44 | 45.81 | 20.82 | 76.54 |
| | | +Ours | **71.37** | **53.19** | **85.29** | **67.57** | **57.92** | **80.55** | **46.77** | **20.94** | **76.83** |
| | PromptSRC | +ProTeCt | 73.07 | 49.54 | 85.16 | 70.61 | 55.52 | 78.73 | 44.56 | 20.36 | 74.42 |
| | | +Ours-Euc | 73.16 | 50.05 | 85.17 | 70.09 | 56.84 | 79.80 | 46.96 | 22.60 | 77.17 |
| | | +Ours-HypV1 | 72.21 | 51.03 | 85.36 | 70.39 | 57.76 | 79.84 | 46.22 | 22.74 | 77.22 |
| | | +Ours-HypV2 | 72.43 | 51.59 | 85.25 | 70.56 | 57.75 | 79.86 | 46.59 | 22.65 | 77.20 |
| | | +Ours | **73.54** | **51.91** | **85.76** | **70.64** | **57.79** | **79.94** | **46.98** | **23.03** | **77.32** |
| 16 | MaPLe | +ProTeCt | 75.34 | 61.15 | 88.04 | 72.17 | 59.71 | 82.27 | 48.14 | 24.82 | 78.79 |
| | | +Ours-Euc | 76.99 | 68.01 | 90.55 | 74.07 | 66.81 | 85.36 | 68.96 | 51.81 | 87.15 |
| | | +Ours-HypV1 | 77.62 | 69.05 | 90.82 | 75.10 | 68.26 | 85.99 | 67.41 | 52.85 | 87.18 |
| | | +Ours-HypV2 | 77.69 | 69.33 | 90.71 | 75.19 | 68.65 | 85.92 | 69.67 | 52.73 | 87.01 |
| | | +Ours | **77.92** | **69.38** | **90.89** | **75.47** | **68.67** | **86.02** | **69.96** | **53.65** | **87.27** |
| | PromptSRC | +ProTeCt | 78.76 | 66.74 | 90.79 | 75.54 | 66.01 | 84.75 | 56.40 | 33.92 | 82.47 |
| | | +Ours-Euc | 78.34 | 68.05 | 90.85 | 75.81 | 68.81 | 86.17 | 66.50 | 48.73 | 87.26 |
| | | +Ours-HypV1 | 78.82 | 68.24 | 91.00 | 76.50 | 69.10 | 86.02 | 67.25 | 49.98 | 87.31 |
| | | +Ours-HypV2 | 78.55 | 68.18 | 91.06 | 76.47 | 69.17 | 86.09 | 66.92 | 49.48 | 87.13 |
| | | +Ours | **78.90** | **68.47** | **91.12** | **76.54** | **69.18** | **86.20** | **67.38** | **50.77** | **87.60** |

## C.4 ABLATION RESULTS ON BASE-TO-BASE/BASE-TO-NOVEL/BASE-TO-WHOLE GENERALIZATION.

We also conduct an ablation study on base-to-base/base-to-novel/base-to-whole generalization settings. The results in Table 7 demonstrate the effectiveness and robustness of our semantic-aware visual feature extraction and intermediate manifold search module.

Table 7: Ablation results for base-to-base/base-to-novel/base-to-whole generalization experiments on SUN.

| Base Method | Variant | LA | | | | HCA | | | | MTA | | | |
|---|---|---|---|---|---|---|---|---|---|---|---|---|---|
| | | Base | Novel | HM | Whole | Base | Novel | HM | Whole | Base | Novel | HM | Whole |
| MaPLe | +ProTeCt | 81.77 | 76.67 | 79.14 | 69.80 | 64.27 | 57.43 | 59.52 | 53.74 | 85.30 | 81.10 | 83.15 | 76.37 |
| | +Ours-Euc | 81.83 | 76.74 | 79.20 | 69.79 | 71.74 | 53.75 | 61.46 | 54.59 | 88.22 | 80.80 | 84.35 | 77.81 |
| | +Ours-HypV1 | 82.22 | 76.90 | 79.47 | 69.68 | 73.00 | 55.02 | 62.75 | 56.64 | 88.42 | 80.80 | 84.44 | 78.16 |
| | +Ours-HypV2 | 82.15 | 76.93 | 79.45 | 69.66 | 73.28 | 55.75 | 63.32 | 56.16 | 88.65 | 80.90 | 84.60 | 78.08 |
| | +Ours | **82.79** | **77.11** | **79.85** | **69.82** | **73.38** | **56.23** | **63.67** | **57.09** | **88.85** | **81.24** | **84.87** | **78.61** |
| PromptSRC | +ProTeCt | 82.36 | 78.40 | 80.33 | 71.94 | 66.86 | 58.80 | 62.57 | 57.06 | 86.67 | 82.76 | 84.67 | 79.12 |
| | +Ours-Euc | 82.85 | 78.83 | 80.79 | 71.93 | 72.28 | 57.69 | 64.17 | 57.51 | 88.74 | 82.65 | 85.59 | 78.93 |
| | +Ours-HypV1 | 83.10 | 78.53 | 80.75 | 72.02 | 72.96 | 58.93 | 65.20 | 57.99 | 88.78 | 82.74 | 85.65 | 79.23 |
| | +Ours-HypV2 | 83.02 | 78.30 | 80.59 | 71.89 | 72.72 | 57.84 | 64.43 | 58.41 | 88.88 | 82.72 | 85.69 | 79.58 |
| | +Ours | **83.40** | **78.72** | **80.99** | **72.04** | **73.11** | **59.10** | **65.36** | **58.42** | **89.02** | **82.86** | **85.83** | **79.74** |

## C.5 ABLATION RESULTS ON MANIFOLD DISTANCE DEFINITION.

We also conduct ablation experiments on our manifold distance definition. Table 8 compares our proposed distance measure with three baselines: squared difference $(c_1 - c_3)^2$, squared log distance $(lnc_1 - lnc_3)^2$, and squared reciprocal distance $(\frac{1}{c_1} - \frac{1}{c_3})^2$. Our method consistently achieves the best performance. These results validate that our distance measure better captures the intrinsic relationships between manifolds with different curvatures compared to simple mathematical transformations.

Table 8: Comparison of different distance metrics for hierarchical learning. Our proposed hyperbolic distance achieves the best performance across all evaluation metrics.

| Method | $D_{\mathcal{L}}(\mathcal{L}^{c_1}, \mathcal{L}^{c_3})$ | LA | HCA | MTA |
|---|---|---|---|---|
| Squared distance | $(c_1 - c_3)^2$ | 71.03 | 50.76 | 84.58 |
| Squared log distance | $(\ln c_1 - \ln c_3)^2$ | 71.10 | 52.42 | 84.95 |
| Squared reciprocal distance | $\left(\dfrac{1}{c_1} - \dfrac{1}{c_3}\right)^2$ | 71.05 | 52.11 | 84.71 |
| Ours | $\dfrac{-\sqrt{c_1} + 2\sqrt{c_3}\cosh[(\sqrt{c_3} - \sqrt{c_1})r]}{2\sqrt{c_1}c_3}$ | **71.37** | **53.19** | **85.29** |

## C.6 ABLATION RESULTS ON THE CROSS-ATTENTION MODULE.

Table 9: Performance of our method without the cross attention module.

| Base Method | Variant | Metrics | | |
|---|---|---|---|---|
| | | LA | HCA | MTA |
| MaPLe | Vanilla | 68.75 | 4.65 | 50.60 |
| | +ProTeCt | 69.33 | 48.10 | 83.36 |
| | +Ours | **71.37** | **53.19** | **85.29** |
| | +Ours w/o cross attention | 67.43 | 48.42 | 84.04 |
| PromptSRC | Vanilla | 72.48 | 14.36 | 51.91 |
| | +ProTeCt | 73.07 | 49.54 | 85.16 |
| | +Ours | **73.54** | **51.91** | **85.76** |
| | +Ours w/o cross attention | 73.12 | 50.11 | 85.08 |

We also conduct an ablation analysis on the cross-attention mechanism by removing it. Specifically, we manually design a set of weighting coefficients for the class tokens at different layers to construct hierarchical visual features. For example, the weight for level 1 is [0.2397, 0.3235, 0.4368], and the weight for level 2 is [0.1628, 0.2966, 0.54051] for training. During inference, given an image and a set of candidate textual labels, we compute visual features at all levels and evaluate their similarities to all textual labels, and the label with the highest similarity is the final prediction. Results are shown in Table 9, highlighting the importance of this mechanism.

## C.7 ADDITIONAL QUANTITATIVE RESULTS FOR LEARNED IMAGE REPRESENTATIONS

Table 10: Clustering performance of our learned representations across taxonomic levels. Our method consistently surpasses the baseline in all three metrics: Silhouette Score and Calinski–Harabasz Index, and Davies–Bouldin Index.

| Taxonomic Level | Silhouette Score ($\uparrow$) | | Calinski-Harabasz Index ($\uparrow$) | | Davies-Bouldin Index ($\downarrow$) | |
|---|---|---|---|---|---|---|
| | Baseline | Ours | Baseline | Ours | Baseline | Ours |
| Phylum | 0.0106 | **0.0112** | 105.7 | **128.0** | 4.48 | **3.78** |
| Class | 0.0191 | **0.0250** | 113.1 | **136.4** | 3.92 | **3.50** |
| Order | 0.0301 | **0.0381** | 48.5 | **56.8** | 3.33 | **3.04** |
| Family | 0.0563 | **0.0671** | 33.8 | **39.5** | 3.24 | **2.99** |
| Genus | 0.0680 | **0.0796** | 27.0 | **31.6** | 3.16 | **2.94** |
| Species | 0.0621 | **0.0727** | 23.4 | **27.3** | 3.38 | **3.19** |

To comprehensively demonstrate that our learned features outperform the baseline across all hierarchical levels, we employ three clustering quality metrics for quantitative comparison: Silhouette Score, Calinski-Harabasz Index, and Davies-Bouldin Index. The Silhouette Score measures how similar a sample is to its own cluster compared to other clusters, with values ranging from -1 to

1 where higher scores indicate better-defined clusters. The Calinski-Harabasz Index evaluates the ratio of between-cluster dispersion to within-cluster dispersion, where higher values signify more distinct and compact clusters. The Davies-Bouldin Index quantifies the average similarity between each cluster and its most similar cluster, with lower values indicating better cluster separation.

Table 10 presents the quantitative comparison between our method and the baseline. The results consistently demonstrate that our hierarchical feature extraction framework produces more discriminative and well-separated representations at each semantic level, indicating that our semantic-aware approach effectively captures the inherent structure of visual concepts at multiple granularities.

## C.8 SENSITIVITY ANALYSIS

Table 11: Performance of our method with different choice of intermediate layers $\{p_j\}_{j=1}^m$ for curvatures of the visual and textual manifolds.

| Intermediate Layers | LA | HCA | MTA |
|---|---|---|---|
| [0,5,11] | 71.08 | 52.52 | 84.65 |
| [2,6,11] | 71.20 | 52.69 | 84.89 |
| [4,7,11] (ours) | **71.37** | **53.19** | **85.29** |
| [6,8,11] | 71.25 | 52.63 | 84.85 |
| [8,9,11] | 71.18 | 52.46 | 84.67 |

Table 12: Sensitivity analysis. (a) Performance with different curvature initialization values. (b) Performance with different weight factors ($\alpha$).

(a)

| Image | Text | LA | HCA | MTA |
|---|---|---|---|---|
| 0.25 | 0.5 | 70.62 | 52.70 | 84.95 |
| 0.5 | 0.25 | 70.88 | 52.95 | 85.02 |
| 0.5 | 0.5 | **71.37** | **53.19** | **85.29** |
| 0.6 | 0.6 | 70.87 | 53.15 | 85.20 |
| 0.7 | 0.7 | 71.18 | 52.72 | 85.04 |

(b)

| $\alpha$ | LA | HCA | MTA |
|---|---|---|---|
| 0.3 | 70.87 | 52.88 | 84.76 |
| 0.4 | 71.02 | 52.90 | 84.90 |
| 0.5 (ours) | **71.37** | **53.19** | **85.29** |
| 0.6 | 70.94 | 53.12 | 85.17 |
| 0.7 | 70.74 | 52.94 | 84.99 |

We conduct experiments to analyze the sensitivity of hyperparameters, including the choice of intermediate transformer layers $\{p_j\}_{j=1}^m$, the initial values for the learnable curvatures, and the loss weighting factor. Specifically, we conduct experiments on the Cifar100 dataset by adjusting these hyperparameters. The results shown in Table 11 and 12, indicating that the method is only minimally sensitive to hyperparameter variation.

Table 13: Performance comparison with different backbone architectures. (a) Results using MaPLe with ViT-L/14 backbone. (b) Results using PromptSRC with ViT-L/14 backbone.

(a)

| Method | LA | HCA | MTA |
|---|---|---|---|
| MaPLe | 75.79 | 10.13 | 58.94 |
| +ProTeCt | 75.54 | 55.90 | 85.89 |
| +Ours | **77.04** | **56.54** | **86.71** |

(b)

| Method | LA | HCA | MTA |
|---|---|---|---|
| PromptSRC | 79.45 | 12.23 | 58.34 |
| +ProTeCt | 79.59 | 57.29 | 86.53 |
| +Ours | **79.99** | **59.01** | **87.39** |

To further investigate the robustness of our method, we additionally utilize another architecture with deeper depths (*i.e.*, CLIP-ViT-L/14) on Cifar100 to conduct experiments on a few-shot setting. Results in Table 13 demonstrate the robustness of our method.

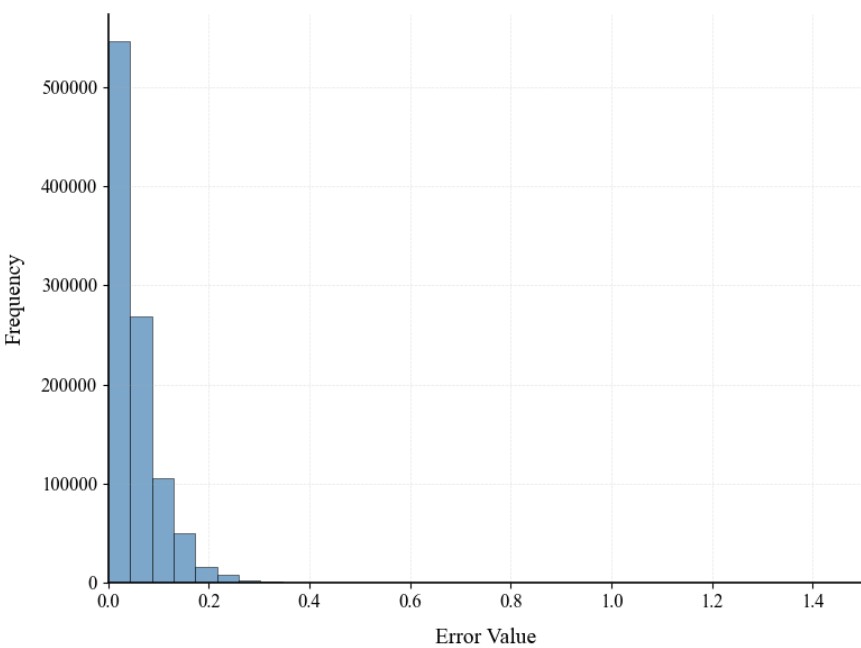

Figure 7: Histogram of the error ratio of Taylor approximation.

### C.9 DISTRIBUTION OF TAYLOR APPROXIMATION ERROR FOR HYPERBOLIC DISTANCE COMPUTATION, SHOWING A MEAN RELATIVE ERROR OF 5.3% ACROSS 1M VECTOR PAIRS.

The Taylor approximation used to compute the KL-based manifold distance is applied to the hyperbolic distance. We conduct an empirical analysis to evaluate this approximation error in Taylor. Specifically, we randomly select 1,000,000 pairs of hyperbolic vectors generated by our method, compute both the exact hyperbolic distance and its Taylor-approximated value, and measure the relative error defined as $error = \frac{|d_{exact} - d_{approx}|}{|d_{exact}|}$. The histogram of this ratio is illustrated in Figure 7. Results show that the mean relative error is 5.3%, with a maximum of 32.4% and a minimum of less than 0.1%, demonstrating that the approximation error is acceptable and controllable in practice.

### C.10 ANALYSIS ON THE ESTIMATION OF $r$

The approximation constant $r$ is the norm of feature midpoints on tangent spaces. In our implementation, $r$ is estimated per batch by computing the mean feature vector on the tangent space and taking its norm. This estimation makes it naturally consistent with the training dynamics—its value evolves smoothly with feature distribution of the model. We compute the relative error defined as $error_r = \frac{|r_{exact} - r|}{|r_{exact}|}$, where $r_{exact}$ is computed on full-batch data. We further visualize the trajectory of $error_r$ during training, as shown in Figure 8

Table 14: Analysis on hyperparameter $r$ in our hyperbolic distance formulation.

| $r$ | LA | HCA | MTA |
|------|-------|-------|-------|
| 1 | 70.58 | 52.41 | 84.48 |
| 2 | 70.52 | 52.46 | 84.50 |
| Ours | **71.37** | **53.19** | **85.29** |

Finally, we set $r$ to a fixed constant, *i.e.*, $r = 1$ and $r = 2$, where results are shown in Table 14. Results demonstrate that our method is robust to the variation of $r$. In conclusion, our method is robust to the approximation of $r$.

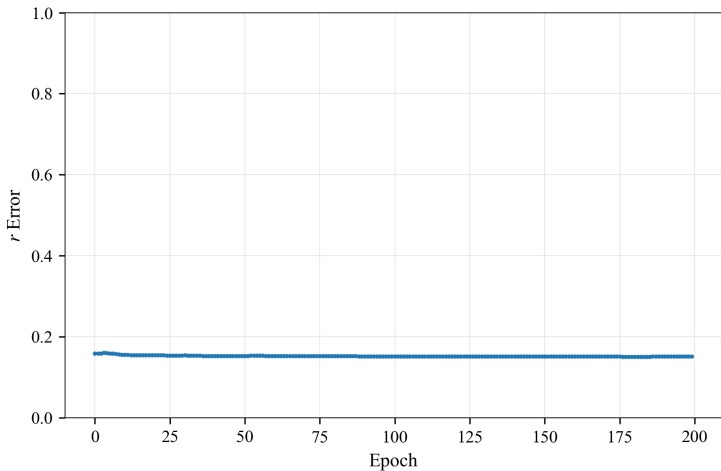

Figure 8: Trajectory of the relative error in estimating $r$.

## C.11 An alternative to addressing species-level ambiguity in Rare Species

Table 15: TOS classification results on Rare Species using an alternative labeling strategy without hierarchical concatenation, under 1-shot and 16-shot settings.

| K Shot | Baseline Method | Variant | Metrics | | |
|---|---|---|---|---|---|
| | | | LA | HCA | MTA |
| 1 | MaPLe | Vanilla | 38.09 | 1.84 | 35.19 |
| | | +ProTeCt | 35.67 | 7.84 | 61.69 |
| | | +Ours | **38.34** | **10.55** | **69.47** |
| | PromptSRC | Vanilla | 39.13 | 6.09 | 43.68 |
| | | +ProTeCt | 37.92 | 9.93 | 64.45 |
| | | +Ours | **39.97** | **12.47** | **71.93** |
| 16 | MaPLe | Vanilla | 50.69 | 2.01 | 35.19 |
| | | +ProTeCt | 44.68 | 16.19 | 61.69 |
| | | +Ours | **66.71** | **44.01** | **77.27** |
| | PromptSRC | Vanilla | 59.99 | 4.55 | 40.54 |
| | | +ProTeCt | 55.53 | 28.66 | 74.76 |
| | | +Ours | **65.16** | **44.43** | **79.74** |

Table 16: Base-to-base/base-to-novel/base-to-whole generalization results on Rare Species using an alternative labeling strategy without hierarchical concatenation.

| Base Method | Variant | LA | | | | HCA | | | | MTA | | | |
|---|---|---|---|---|---|---|---|---|---|---|---|---|---|
| | | Base | Novel | HM | Whole | Base | Novel | HM | Whole | Base | Novel | HM | Whole |
| MaPLe | Vanilla | 50.38 | 39.29 | 44.15 | 41.09 | 3.20 | 2.90 | 3.04 | 1.84 | 8.51 | 8.15 | 8.33 | 7.42 |
| | +ProTeCt | 45.50 | 38.80 | 41.88 | 37.92 | 21.78 | 8.44 | 12.17 | 10.97 | 18.69 | 17.45 | 18.05 | 18.14 |
| | +Ours | **66.86** | **40.40** | **50.37** | **47.77** | **50.97** | **10.10** | **16.86** | **24.57** | **19.15** | **20.01** | **19.57** | **20.07** |
| PromptSRC | Vanilla | 60.13 | 42.20 | 49.59 | 46.43 | 6.06 | 6.04 | 6.05 | 4.01 | 7.64 | 8.00 | 7.82 | 6.25 |
| | +ProTeCt | 54.33 | 42.02 | 47.39 | 44.93 | 33.22 | 11.75 | 17.36 | 17.36 | 18.08 | 16.12 | 17.04 | 17.88 |
| | +Ours | **64.93** | **44.21** | **52.60** | **48.85** | **48.36** | **12.09** | **19.34** | **24.74** | **18.92** | **16.72** | **17.75** | **18.42** |

We additionally design an alternative approach that combines the common name with the species name to resolve ambiguity at the species level. Specifically, we replace non-unique labels with a

combination of the scientific name and the common name to guarantee uniqueness without relying on the concatenated hierarchy. We re-conduct experiments without using the concatenation strategy. Results in Table 15 and 16 show that our method significantly improves the performance of the compared methods, demonstrating its superiority. Notably, in the 16-shot setting of the few-shot task, our method achieves up to a 22.03% improvement on LA, a 27.82% improvement on HCA, and a 15.58% improvement on MTA, indicating its ability to align the textual and visual modalities with the hierarchical semantic structures.

## C.12 POTENTIAL DATA IMBALANCE.

Table 17: Analysis on class imbalance using the MaPLe baseline.

| Method | LA | HCA | MTA |
|---|---|---|---|
| MaPLe | 68.75 | 4.65 | 50.60 |
| + ProTeCt | 69.33 | 48.10 | 83.36 |
| + Ours | **71.37** | 53.19 | 85.29 |
| + Ours-balanced | 71.26 | **53.54** | **85.70** |

Class imbalance is a common issue. This issue is particularly prevalent in taxonomic open-set classification tasks. At the leaf level, sample sizes per category are typically balanced, whereas at higher taxonomic levels, the distribution can be highly imbalanced. We note that CLIP has already shown the capability to handle such an imbalance issue (Wen et al., 2024). Therefore, we did not explicitly consider the imbalance issue in our current design. In future work, we plan to develop more robust algorithms to address this issue. As a preliminary attempt, we have implemented a sample re-weighting strategy to mitigate potential bias. As shown in Table 17, this balanced approach brings consistent improvements in hierarchical metrics (HCA and MTA). We attribute these improvements to the fact that the re-weighting strategy effectively addresses the imbalance at non-leaf nodes, allowing the model to better learn the hierarchical structure of the data.

## C.13 MORE VISUALIZATIONS ON RARE SPECIES

We visualize the attention maps generated by the baseline ProTeCt (Wu et al., 2024) and our model using GradCAM (Selvaraju et al., 2017) to analyze their behaviors across different taxonomic levels. The results are shown in Figures 9 and 10. When aligned with text prompts at different granularities, our model exhibits hierarchical attention patterns, focusing on distinct visual regions that are most discriminative for each taxonomic level. Specifically, we observe the following attention patterns:

- **Giant Panda (Ailuropoda melanoleuca):** At the *class* level (Mammalia), the model primarily attends to the distinctive fur texture and body shape, which are key features distinguishing mammals from other vertebrate classes (*e.g.*, Chondrichthyes). At the *order* level (Carnivora), attention shifts to the limbs and paws, capturing the characteristic plantigrade locomotion that differentiates carnivorans from other mammalian orders (*e.g.*, Primates). At the *genus* level (Ailuropoda), the model focuses on the facial features and distinctive black-and-white coloration pattern, which uniquely identifies giant pandas from other bear genera (*e.g.*, Tremarctos).

- **Forty-Spotted Pardalote (Pardalotus quadragintus):** At the *class* level (Aves), the model attends to the head plumage and feather structure, fundamental avian characteristics that distinguish birds from other vertebrates. At the *order* level (Passeriformes), attention concentrates on the bill morphology, particularly its short and stubby shape that characterizes pardalotes and differentiates them from other passerines with elongated bills (*e.g.*, honeyeaters with long, slender bills) or from non-passerine orders with distinct bill structures (*e.g.*, Psittaciformes with curved beaks). At the *family* level (Pardalotidae), the model highlights the distinctive wing patterns, specifically the characteristic spotted markings that distinguish pardalotes from other passerine families within the same order.

- **Grey Reef Shark (Carcharhinus amblyrhynchos):** At the *class* level (Chondrichthyes), the model focuses on the caudal fin structure, a defining feature of cartilaginous fishes

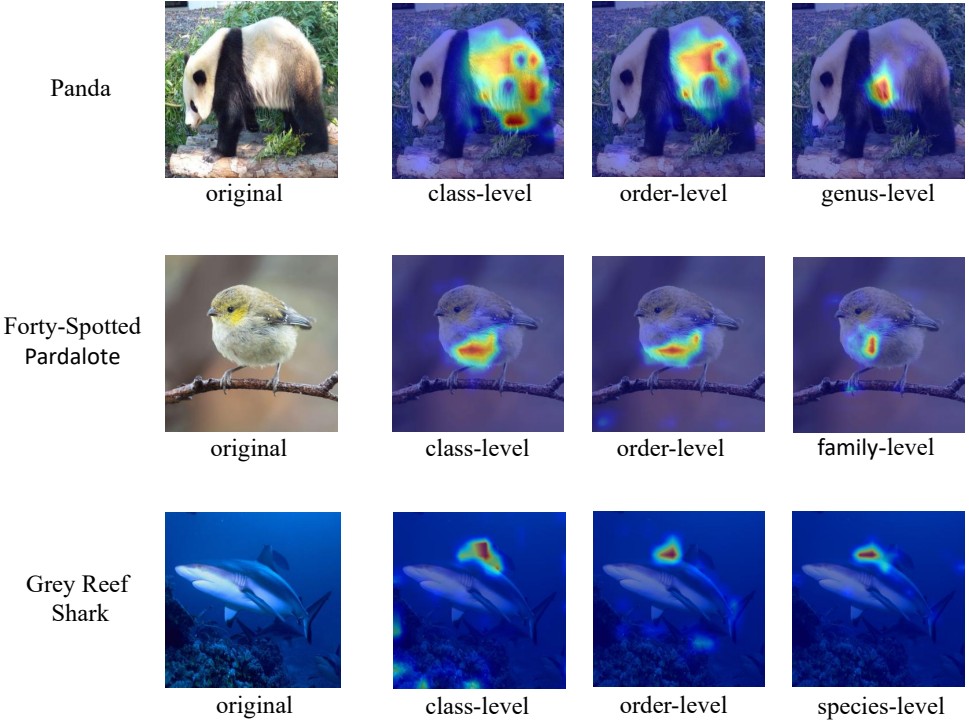

Figure 9: Visualization of attention maps generated by the baseline ProTeCt (Wu et al., 2024) on Rare Species.

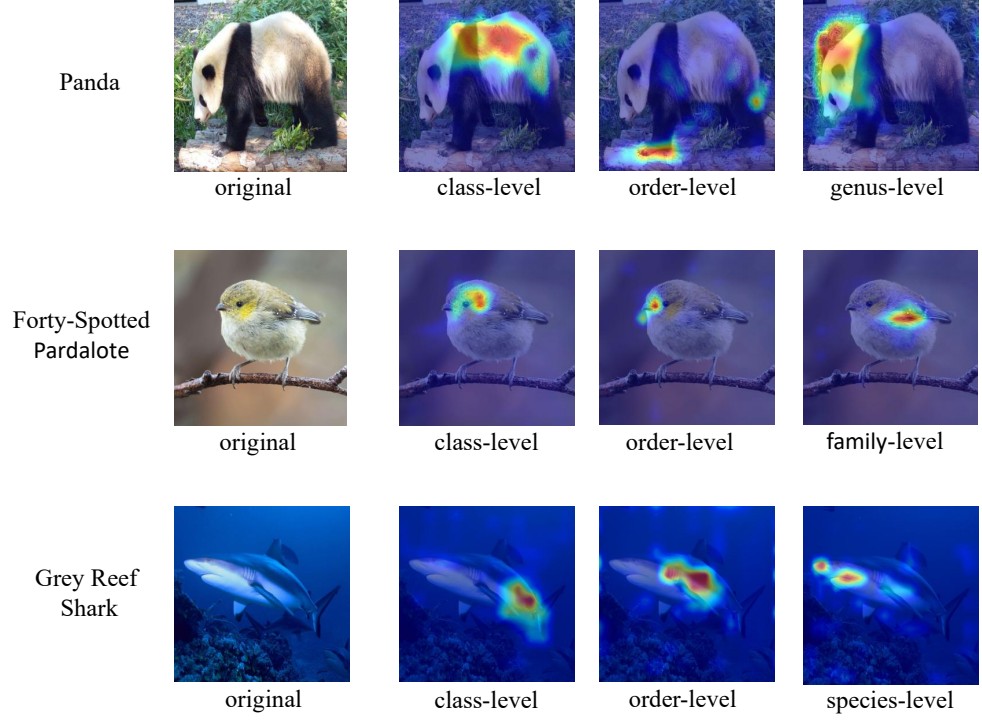

Figure 10: Visualization of attention maps generated by our model on Rare Species.

that distinguishes them from bony fishes (Osteichthyes). At the *order* level (Carcharhiniformes), attention is directed towards the gill region, specifically the presence of five gill slits—a diagnostic feature of ground sharks that differentiates them from other shark orders (*e.g.*, Hexanchiformes with six or seven gill slits). At the *species* level (Amblyrhynchos), the model identifies the snout morphology and mouth position, which are species-specific characteristics distinguishing the grey reef shark from other Carcharhinus species.

These visualization results demonstrate that our model learns biologically meaningful features at each taxonomic level, aligning with domain knowledge in taxonomy and supporting its strong performance on hierarchical classification tasks.

## C.14 VISUALIZATION FOR ARTIFACT CLASSES.

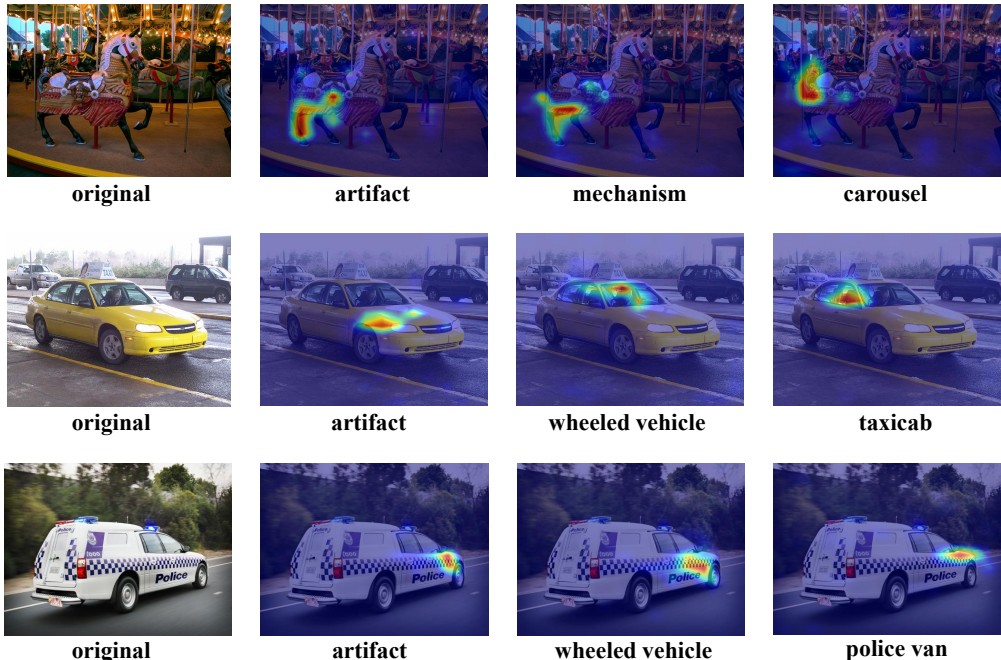

Figure 11: Visualization of attention maps generated by the baseline ProTeCt (Wu et al., 2024) on ImageNet.

We also visualize the attention maps generated by our model and baseline when encountering artifact classes. The results are presented in Figures 11 and 12. When processing text descriptions at different granularities, our model demonstrates adaptive attention to semantically relevant regions. For instance, when determining whether images of a taxicab or a police van belong to the wheeled vehicle category, our model correctly attends to the wheel region, whereas the baseline model fails to focus on this discriminative feature. This further demonstrates that our semantic-aware visual feature extraction framework effectively leverages textual cues to guide attention towards hierarchically-appropriate visual regions, enabling more precise alignment between coarse-to-fine semantics across modalities.

## C.15 LEARNED CURVATURES ON DIFFERENT DATASETS.

We additionally provide the learned and intermediate curvatures across different datasets, with results shown in Table 18. We observe that the curvatures of the text, image, and intermediate manifolds differ substantially for each dataset, indicating that data-driven curvature learning is necessary to match the varying geometric structures. The large divergence among the three learned curvatures further confirms that different modalities indeed require different curvature adaptations, validating the core motivation of our method.

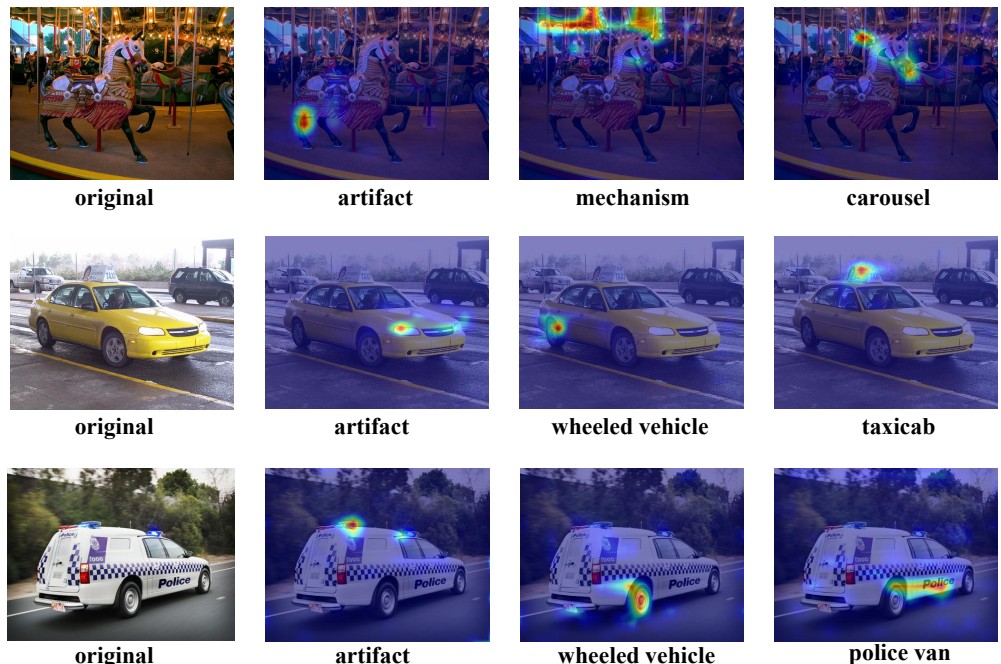

Figure 12: Visualization of attention maps generated by our model on ImageNet.

Table 18: Learned curvatures and the corresponding optimal curvatures for intermediate manifolds of our method on different datasets.

| Dataset | Images | Text | Intermediate |
|---|---|---|---|
| Cifar100 | 0.11 | 0.37 | 0.26 |
| SUN | 0.019 | 0.029 | 0.026 |
| ImageNet | 0.022 | 0.028 | 0.026 |
| Rare Species | 0.15 | 0.40 | 0.30 |

## C.16 EFFICIENCY ANALYSIS

We evaluate the efficiency of our method on the Rare Species dataset, using batch sizes of 8 for training and 1024 for inference. We measure the per-batch time cost and memory usage, reporting results averaged over 1,000 batches as shown in Figure 13. The results demonstrate that our method introduces only modest overhead—approximately 30% more time and 2% more GPU memory during training, and 10% more time and 4% more GPU memory during inference. Moreover, learning multiple curvatures introduces almost no additional cost.

## D    MORE DISCUSSIONS

### D.1    THEORETICAL JUSTIFICATION FOR THE SEMANTIC-AWARE FEATURE EXTRACTION MODULE

In this section, we provide a detailed theoretical justification for the design of our semantic-aware feature extraction module, specifically addressing the rationale behind layer selection and the suppression of cross-token attention. We further discuss the robustness and novelty of this module compared to existing approaches.

Our strategy of selecting specific intermediate transformer layers is grounded in the Information Bottleneck (IB) theory applied to deep networks (Tishby & Zaslavsky, 2015). As network depth increases, representations undergo a progressive compression process that reduces the mutual infor-

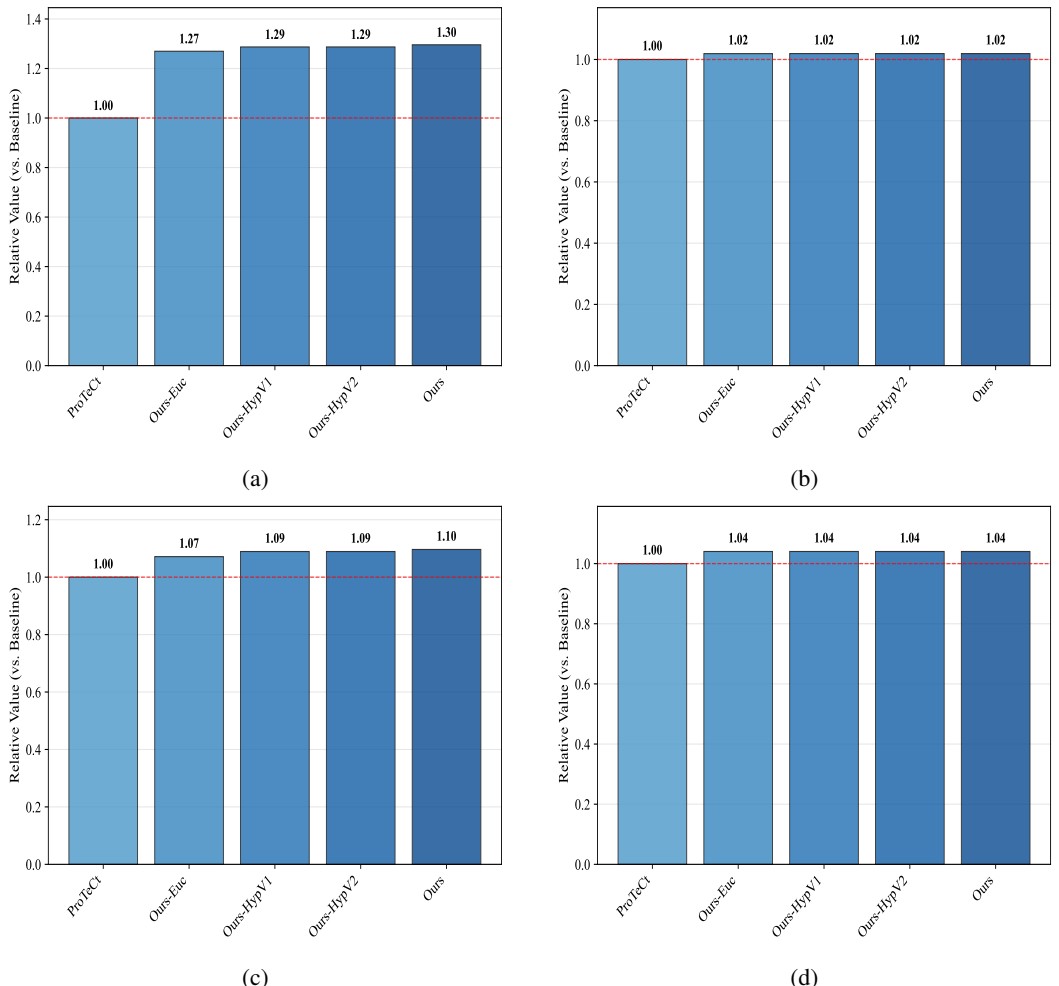

Figure 13: Efficiency analysis of different methods. (a) Training time consumption; (b) Training memory consumption; (c) Inference time consumption; (d) Inference memory consumption.

mation between the features and the input. This compression induces a natural hierarchy of semantic granularity:

- Early layers capture coarse and low-level visual cues.
- Intermediate layers can model mid-level semantic representations.
- Late layers are highly compressed and encode fine-grained semantics.

This theoretical progression motivates our coarse-to-fine feature extraction strategy. While we select layers to align with these distinct granularities, we empirically demonstrate that the specific choice of layers within these ranges does not significantly affect performance (see Table 11 in Appendix C.8).

From a theoretical standpoint, the self-attention mechanism operates as a message-passing or kernel-smoothing operator (Cordonnier et al., 2020; Tsai et al., 2019). This mechanism inevitably mixes information across tokens and induces semantic smoothing at each scale (Park & Kim, 2022). Such cross-token mixing hinders the preservation of specific granularity semantics because the feature of each token becomes entangled with information from other spatial regions. Consequently, the visual evidence aggregated into the class token may no longer align with the intended semantic granularity of that layer. Motivated by these findings, we disable cross-token attention during feature extraction to ensure that the class token summarizes disentangled, semantically coherent features at each granularity, which is crucial for robust multi-modal alignment.

To further verify that the effectiveness of our component is not sensitive to specific architectures (e.g., varying depths of ViT), we conducted additional experiments using a larger backbone, CLIP-ViT-L/14, on the Cifar100 few-shot setting. As shown in Table 13 in Appendix C.8, our method achieves consistent performance gains over baselines (e.g., MaPLe and ProTeCt) on the deeper backbone. These results demonstrate the generalizability and robustness of our core component across different model capacities.

While utilizing intermediate features is a well-established technique, our semantic-aware module differs significantly from existing methods that typically operate in a purely feed-forward manner (e.g., weighted fusion or concatenation). Our approach introduces two key innovations:

- We introduce cross-token attention suppression to construct features that strictly preserve coarse-to-fine semantics, a strategy that has not been explored in existing feature extraction methods.
- We design a semantic-aware module that utilizes textual semantics to actively guide the coarse-to-fine visual feature construction.

Moreover, our contribution extends beyond feature extraction. We align these hierarchical, tree-like semantic features across heterogeneous manifolds, ensuring that the similarity in using intermediate layers does not diminish the novelty of our overall framework.

## D.2 How does our model avoid degenerate solutions?

Our model avoids the degeneration of fine-grained features (*i.e.*, preventing fine-grained features from collapsing into coarser representations) through three synergistic mechanisms:

- **Feature construction preserves fine-level detail.** We explicitly include the final-layer class token to construct visual representations. Specifically, we utilize the finest-grained textual feature as the query to compute the attention weights on candidate tokens. Empirically, the average attention weights on layers 4, 7, and 11 are 0.06, 0.08, and 0.86, respectively. We observe that the attention weights on this final-layer token are significantly larger than those on intermediate layers. This indicates that the finest-grained features successfully capture fine-grained details and do not collapse into coarser representations.
- **Fine-grained alignment loss directly prevents collapse.** Our training objective $J_{\text{pro}}(T_e, V_e)$ in Eq.(17) explicitly includes a distance alignment loss between the finest-level textual and visual features. This objective forces the model to maintain a discriminative fine-grained structure, as any degeneration would inevitably increase this alignment loss.
- **Modality-specific geometric constraints enforce boundary separation.** The constraints defined in Eq.(15) encourage fine-grained features to reside near the boundary of the hyperbolic manifolds. Because points near the boundary in hyperbolic space have larger radii and correspond to more specific semantics, our method naturally prevents fine-level features from collapsing to coarser levels.

## E The Use of Large Language Models (LLMs)

In this work, we employed Large Language Models (LLMs) as assistive tools in the following capacities:

**Code Development:** We utilized LLMs to help understand existing codebases and assist in writing new code implementations. Specifically, LLMs were used for:

- Explaining complex code segments and algorithmic implementations
- Generating code snippets and suggesting optimizations
- Debugging and identifying potential issues in our implementations

**Writing Assistance:** LLMs were employed to improve the clarity and readability of our manuscript through:

- Refining technical descriptions and explanations
- Improving grammar and language flow
- Suggesting alternative phrasings for better clarity

**Limitations of LLM Usage:** We emphasize that:

- All research ideas, experimental design, and core contributions are original work by the authors
- All LLM-generated content was carefully reviewed, verified, and modified by the authors
- The experimental results and analysis are entirely based on our own implementations and observations

The specific LLMs used include ChatGPT, Claude, and Deepseek-R1.

