# OpenReview forum: "Modality Alignment across Trees on Heterogeneous Hyperbolic Manifolds"
_ICLR.cc/2026/Conference — ICLR 2026 Poster_

### Official Review · Reviewer_8Seg · 2025-10-27

**Soundness:** 3
**Presentation:** 3
**Contribution:** 3
**Rating:** 6
**Confidence:** 3

**Summary:**

This paper proposes "Alignment across Trees," a novel method for aligning vision and language modalities by constructing symmetric, hierarchical feature representations for both. The core idea is to address the asymmetry in prior work where hierarchical text features are aligned with a single, flat image feature. The method introduces two main contributions: 1) a semantic-aware visual feature extraction framework that uses cross-attention with textual cues to build a hierarchical tree of visual features from intermediate Transformer layers, and 2) a heterogeneous manifold alignment algorithm that embeds the image and text feature trees into separate hyperbolic manifolds with distinct, learnable curvatures. To align these, the paper defines a novel distance measure between heterogeneous hyperbolic manifolds and learns an optimal intermediate manifold. The approach is evaluated on taxonomic open-set classification, where it shows consistent and significant improvements over strong baselines.

**Strengths:**

1. The paper clearly identifies a key limitation in existing VLMs, asymmetric feature representation, and proposes a well-motivated solution.

2. The use of heterogeneous hyperbolic manifolds with learnable curvatures to capture the intrinsic geometry of each modality is technically sophisticated. The formulation of an intermediate manifold for alignment, supported by proofs of existence and uniqueness, provides a strong theoretical foundation. Even though I do not have enough experience in this line of work to confirm with certainty the correctness of the proofs.

3. The method demonstrates substantial and consistent performance gains over strong baselines across multiple datasets, metrics, and experimental settings (few-shot and generalization).

**Weaknesses:**

1. The paper’s literature review, while good, overlooks several relevant lines of work. It would be stronger if it discussed the concept of emergent alignment in models, and positioned the proposed "intermediate manifold" with respect to related concepts such as:
    - The idea of a universal or shared latent space, as explored in work on relative representations [1,2]. The proposed "intermediate manifold" seems to be a generalization of this concept to heterogeneous hyperbolic spaces.
    - The general problem of identifiability in representation learning (e.g., [3]), as the proposed work is implicitly trying to learn an identifiable hierarchical structure


2.  Some of the visual results used to support the paper's claims are difficult to interpret without further evidence. For instance, the qualitative improvement in clustering in Figure 5 is not immediately obvious, and the attention map visualizations in Figure 6 lack a crucial comparison to the baseline.

3. While the paper includes ablations, further analysis would strengthen the claims. For example, an ablation on the novel manifold distance definition and an analysis of the learned curvatures would provide deeper insight into the method's behavior.

---

[1] Moschella, et al, "Relative representations enable zero-shot latent space communication," in ICLR, 2023.

[2] Cannistraci, et al, "From Bricks to Bridges: Product of Invariances to Enhance Latent Space Communication," in ICLR, 2024.

[3] Kivva, et al, "Identifiability of deep generative models without auxiliary information," NeuriIPS 2022.

**Questions:**

1. The manuscript introduces a new function D to measure the distance between hyperbolic manifolds. Is this function a formal distance metric (i.e., does it satisfy non-negativity, identity of indiscernibles, symmetry, and the triangle inequality)? A brief discussion on this would be helpful.

2. The learnable curvatures are an interesting component. Could the authors provide some analysis on the learned values? For instance, do textual and visual manifolds for semantically similar/distinct data end up with correspondingly similar/distinct curvatures? This could provide valuable insight.

3.  Figure 5: The claim that the method achieves "improved feature separability" is hard to verify by visual inspection alone, especially for the "Order" and "Class" levels. Could it be supplemented with a quantitative clustering metric (e.g., Silhouette Score or Calinski-Harabasz Index) to more formally support this claim?

4. Figure 6: These attention maps are very compelling. However, to demonstrate that the method is responsible for this hierarchical attention, it is essential to show the corresponding attention maps from the baseline. Does the regularization and alignment strategy actually cause the model to shift its attention in this structured way compared to the baseline?

5. The abstract mentions that existing methods represent each image with a "single feature." Models like ViT produce patch-level features. What is the final, single-vector representation (e.g., the [CLS] token) used for alignment?

---

> ### Author Response · Authors · 2025-11-22
>
> Thanks for your insightful and thorough review. We will address your concerns one by one.
>
> > **W1:** The paper’s literature review, while good, overlooks several relevant lines of work. It would be stronger if it discussed the concept of emergent alignment in models, and positioned the proposed "intermediate manifold" with respect to related concepts such as: (1) The idea of a universal or shared latent space, as explored in work on relative representations [1,2]. The proposed "intermediate manifold" seems to be a generalization of this concept to heterogeneous hyperbolic spaces.
> The general problem of identifiability in representation learning (e.g., [3]), as the proposed work is implicitly trying to learn an identifiable hierarchical structure.
>
> **R:** Thanks for your suggestions. We have added the following content to the Related Work section.
>
> Some works investigate relative representations that preserve relational geometry, enabling communication between different latent spaces without mapping them into a single shared space. For example, Moschella et al. (ICLR 2023) demonstrate that pairwise relational structure enables information transfer across embedding spaces, and Cannistraci et al. (ICLR 2024) extend this principle through invariance‑preserving transforms. Different from existing methods that model data on Euclidean space and use latent spaces, our method utilizes hyperbolic manifolds for modeling and constructs an intermediate manifold for communication.
>
>
> Moreover, some works implicitly try to learn an identifiable hierarchical structure. For example, Kivva et al. (NeurIPS 2022) study identifiability in deep generative models, and Kong et al. (NeurIPS 2024) explore latent hierarchies by introducing structured discrete variables that capture hierarchical dependencies. However, these methods align hierarchical textual features with a single visual representation, leading to suboptimal performance and ignoring the geometric structure of hierarchical multimodal data.  In contrast, our method extracts hierarchical visual features and models them in hyperbolic manifolds for improved alignment.
>
>
> Moschella L, et al. Relative representations enable zero-shot latent space communication[J]. ICLR 2023.
>
> Cannistraci I, et al. From bricks to bridges: Product of invariances to enhance latent space communication[J]. ICLR 2024.
>
> Kivva B, et al. Identifiability of deep generative models without auxiliary information[J]. NeurIPS, 2022, 35: 15687-15701.
>
> Kong L, et al. Learning discrete concepts in latent hierarchical models[J]. NeurIPS, 2024, 37: 36938-36975.

---

> ### Author Response · Authors · 2025-11-22
>
> > **W2:** Some of the visual results used to support the paper's claims are difficult to interpret without further evidence. For instance, the qualitative improvement in clustering in Figure 5 is not immediately obvious, and the attention map visualizations in Figure 6 lack a crucial comparison to the baseline.
> **Q3:** Figure 5: The claim that the method achieves "improved feature separability" is hard to verify by visual inspection alone, especially for the "Order" and "Class" levels. Could it be supplemented with a quantitative clustering metric (e.g., Silhouette Score or Calinski-Harabasz Index) to more formally support this claim?
> **Q4:** Figure 6: These attention maps are very compelling. However, to demonstrate that the method is responsible for this hierarchical attention, it is essential to show the corresponding attention maps from the baseline. Does the regularization and alignment strategy actually cause the model to shift its attention in this structured way compared to the baseline?
>
>
>
> **R:** Thanks for your suggestions. We compute quantitative clustering metrics ( Silhouette Score, Calinski-Harabasz Index, and Davies-Bouldin Index) of our method and baseline to support our claim of "improved feature separability".  Results in Table 1 support our claim. Analysis has been added to the revised manuscirpt.
>
> **Table 1(a): Silhouette Score(↑) of learned image representations.**
> |          |   Phylum   |   Class   |    Order   |   Family   |    Genus   |   Species  |
> |:--------:|:----------:|:---------:|:----------:|:----------:|:----------:|:----------:|
> | Baseline |   0.0106   |   0.0191  |   0.0301   |   0.0563   |    0.068   |   0.0621   |
> |   Ours   | **0.0112** | **0.025** | **0.0381** | **0.0671** | **0.0796** | **0.0727** |
>
> **Table 1(b): Calinski-Harabasz Index(↑) of learned image representations.**
> |          |   Phylum  |   Class   |   Order  |  Family  |   Genus  |  Species |
> |:--------:|:---------:|:---------:|:--------:|:--------:|:--------:|:--------:|
> | Baseline |   105.7   |   113.1   |   48.5   |   33.8   |   27.0   |   23.4   |
> |   Ours   | **128.0** | **136.4** | **56.8** | **39.5** | **31.6** | **27.3** |
>
> **Table 1\(c\): Davies-Bouldin Index(↓) of learned image representations.**
> |          |  Phylum  |  Class   |   Order  |  Family  |   Genus  |  Species |
> |:--------:|:--------:|:--------:|:--------:|:--------:|:--------:|:--------:|
> | Baseline |   4.48   |   3.92   |   3.33   |   3.24   |   3.16   |   3.38   |
> |   Ours   | **3.78** | **3.50** | **3.04** | **2.99** | **2.94** | **3.19** |
>
> We also conduct attention map visualizations of the baseline method, which have been added to the revised manuscript (Figure 6). Results indicate that our method can attend to different regions corresponding to each taxonomic granularity, while the baseline method only attends to a similar region. Because our method constructs coarse-to-fine visual features and designs a symmetric alignment strategy, our method can attend to different regions corresponding to coarse-to-fine granularities.

---

> ### Author Response · Authors · 2025-11-22
>
> > **W3:** While the paper includes ablations, further analysis would strengthen the claims. For example, an ablation on the novel manifold distance definition and an analysis of the learned curvatures would provide deeper insight into the method's behavior.
> **Q2:** The learnable curvatures are an interesting component. Could the authors provide some analysis on the learned values? For instance, do textual and visual manifolds for semantically similar/distinct data end up with correspondingly similar/distinct curvatures? This could provide valuable insight.
>
> **R:** Thanks for your suggestions. We add an ablation on the manifold distance. We replace the proposed distance with the squared distance, squared log distance, and squared reciprocal distance of curvatures. We evaluate their performance on CIFAR-100.  Results show that our manifold distance consistently yields better performance, demonstrating its advantage in capturing the underlying geometry.
>
> **Table 2: Ablation on the manifold distance.**
> ||$D_{\mathcal{L}}( \mathcal{L}^{c_{1}}, \mathcal{L}^{c_{3}})$|     LA    |    HCA    |    MTA    |
> |:---------:|:----------------------------------------------------------:|:---------:|:---------:|:---------:|
> | squared distance |       $(c_1-c_3)^2$                                        |   71.03   |   50.76   |   84.58   |
> | squared log distance |     $(lnc_1-lnc_3)^2$                                      |   71.10   |   52.42   |   84.95   |
> |squared reciprocal distance | $(\frac{1}{c_1} - \frac{1}{c_3})^2$                        |   71.05   |   52.11   |   84.71   |
> | ours |              $\frac{-\sqrt{c_1}+2\sqrt{c_3}\cosh[(\sqrt{c_3}-\sqrt{c_1})r]}{2\sqrt{c_1}c_3}$                                | **71.37** | **53.19** | **85.29** |
>
>
> We additionally report the learned and intermediate curvatures across different datasets. We observe that the curvatures of the text, image, and intermediate manifolds differ substantially for each dataset, indicating that data-driven curvature learning is necessary to match the varying geometric structures. The large divergence among the three learned curvatures further confirms that different modalities indeed require different curvature adaptations, validating the core motivation of our method.
>
> **Table 3: Learned curvatures on different datasets.**
> |             | images |  Text | intermediate |
> |-------------|:------:|:-----:|:------------:|
> |   Cifar100  |  0.11  |  0.37 |     0.26     |
> |     Sun     |  0.019 | 0.029 |     0.026    |
> |   Imagenet  |  0.022 | 0.028 |     0.026    |
> | Rarespecies |  0.15  |  0.40 |     0.30     |
>
> > **Q1:** The manuscript introduces a new function D to measure the distance between hyperbolic manifolds. Is this function a formal distance metric (i.e., does it satisfy non-negativity, identity of indiscernibles, symmetry, and the triangle inequality)? A brief discussion on this would be helpful.
>
> **A:** Thanks for your suggestions. The proposed distance is derived from the KL divergence, which does not satisfy symmetry or the triangle inequality; consequently, the proposed distance does not satisfy these properties either. In the future, we consider leveraging JS divergence to derive the formal distance metric between manifolds. We have added this discussion to the revised manuscript.
>
>
>
>
> > **Q5:** The abstract mentions that existing methods represent each image with a "single feature." Models like ViT produce patch-level features. What is the final, single-vector representation (e.g., the [CLS] token) used for alignment?
>
> **A:** Although ViT produces many patch features, existing methods such as CLIP use only the [CLS] token as the image representation for cross-modal alignment.  Therefore, we claim that existing methods represent each image with a "single feature" for alignment.

---

### Official Review · Reviewer_1GsE · 2025-10-29

**Soundness:** 3
**Presentation:** 3
**Contribution:** 3
**Rating:** 6
**Confidence:** 3

**Summary:**

This paper tackles the modality gap in hierarchical vision-language learning. The key idea is to construct symmetric tree-like features for both images and texts, and then align them on hyperbolic manifolds. Specifically, the method extracts hierarchical visual features from intermediate transformer layers using a text-guided attention module. To handle geometric differences between modalities, it embeds them into separate hyperbolic spaces with learnable curvatures and aligns them through an optimally constructed intermediate manifold. The approach demonstrates strong performance on taxonomic open-set classification across several datasets.

**Strengths:**

- The paper addresses an important and interesting problem of achieving efficient and symmetric modality alignment for hierarchical semantic structures.
- The work presents a novel approach by leveraging hyperbolic spaces with learnable curvatures and an intermediate manifold for modality alignment between images and text.
- Comprehensive experiments demonstrate the method's effectiveness, showing consistent improvements across multiple datasets and settings for taxonomic classification.

**Weaknesses:**

- The design of the semantic-aware feature extraction module appears somewhat heuristic and lacks strong theoretical justification. The critical choices of which specific intermediate transformer layers to use and the decision to disable cross-token attention are based on empirical observations rather than derived from first principles. This raises concerns about the generalizability and robustness of this core component, as its effectiveness might be sensitive to the specific architecture (e.g., different ViT depths) or pre-training dataset. Furthermore, the utilization of intermediate features is a well-established technique, diminishing the perceived novelty of this part of the proposed framework.
- Concern about sensitivity to hyperparameters. The proposed method introduces a non-trivial number of hyperparameters, and its performance is likely sensitive to their careful tuning. Key hyperparameters include the choice of intermediate transformer layers, the initial values for the learnable curvatures, and the loss weighting factor α. The paper would be strengthened by a sensitivity analysis demonstrating the robustness of the results to variations in these critical settings. Without such an analysis, it is difficult to assess the true practicality and generalizability of the approach, as the reported strong performance might be contingent on a specific, finely-tuned configuration.

**Questions:**

Please kindly refer to the weakness section.

---

> ### Author Response · Authors · 2025-11-22
>
> Thanks for your insightful and thorough review. We will address your concerns one by one.
>
> > **W1:** The design of the semantic-aware feature extraction module appears somewhat heuristic and lacks strong theoretical justification. The critical choices of which specific intermediate transformer layers to use and the decision to disable cross-token attention are based on empirical observations rather than derived from first principles. This raises concerns about the generalizability and robustness of this core component, as its effectiveness might be sensitive to the specific architecture (e.g., different ViT depths) or pre-training dataset. Furthermore, the utilization of intermediate features is a well-established technique, diminishing the perceived novelty of this part of the proposed framework.
>
> **R:** The choices of specific intermediate transformer layers can be supported by the information bottleneck (IB) theory. According to the IB theory on deep networks [1], as depth increases, representations undergo a progressive compression process that reduces the mutual information between the features and the input. This compression induces a natural semantic granularity hierarchy.
> - Early layers capture coarse and low-level visual cues.
> - Intermediate layers can model mid-level semantic representations.
> - Late layers are highly compressed and encode fine-grained semantics.
>
> This theoretical progression motivates our coarse-to-fine feature extraction strategy, i.e., we select early layers for coarse cues, mid layers for mid-level features, and late layers for fine-grained semantics. The specific choice of which layer does not significantly affect the performance, which is empirically demonstrated in the response of W2.
>
>
>
> From a theoretical standpoint, attention operates as a message-passing or kernel-smoothing operator [2,3], which inevitably mixes information across tokens and induces semantic smoothing at each scale [4]. Such cross-token mixing hinders the preservation of current-granularity semantics, because the feature of each token becomes entangled with information coming from other spatial regions. As a result, the visual evidence that should be aggregated into the [CLS] token is no longer aligned with the intended semantic granularity of that layer, leading to degraded hierarchical alignment. Motivated by the findings in [1], we disable cross-token attention during feature extraction to ensure that the [CLS] token summarizes the disentangled, semantically coherent features at each granularity, which is crucial for robust multi-modal alignment.
>
>
>
> To further investigate the robustness of our method, we additionally utilize another architecture with deeper depths (i.e., CLIP-ViT-L/14) on CIFAR100 to conduct experiments on a few-shot setting. Results demonstrate the robustness of our method.
>
> **Table 1(a): Results using MaPLe with VIT-L/14 Backbone.**
> |          | LA        | HCA       | MTA       |
> |----------|-----------|-----------|-----------|
> | MaPLe    | 75.79     | 10.13     | 58.94     |
> | +ProTeCt | 75.54     | 55.90     | 85.89     |
> | +ours    | **77.04** | **56.54** | **86.71** |
>
>
> **Table 1(b): Results using PromptSRC with VIT-L/14 Backbone**
> |          | LA        | HCA       | MTA       |
> |----------|-----------|-----------|-----------|
> | MaPLe    | 79.45     | 12.23     | 58.34     |
> | +ProTeCt | 79.59      | 57.29     | 86.53     |
> | +ours    | **79.99** | **59.01** | **87.39** |
>
>
>
> The key idea of using intermediate features is common, but the semantic-aware module in our method differs from existing methods. Existing methods typically extract features from multiple intermediate layers and construct visual representations through weighted fusion, concatenation, or selective usage, operating in a purely feed-forward manner.
> - We introduce cross-token attention suppression to construct features that preserve coarse-to-fine semantics, which has not been explored in existing feature extraction methods.
> - We design a semantic-aware module that utilizes the text semantics to guide coarse-to-fine visual feature construction, which has not been explored in existing methods.
>
> Moreover, our contribution is not limited to using intermediate layers; rather, we construct hierarchical, tree-like semantic features for both images and texts and align such tree-like features to enhance modality alignment. We additionally introduce other unexplored innovations, such as alignment across heterogeneous manifolds. Therefore, this similarity in using intermediate features does not diminish the novelty or significance of our method.

---

> ### Author Response · Authors · 2025-11-22
>
> Due to the word limit, the list of references is provided separately.
>
> [1] Tishby N, Zaslavsky N. Deep learning and the information bottleneck principle[C]//2015 ieee information theory workshop (itw). Ieee, 2015: 1-5.
>
> [2] Cordonnier J B,et al. On the Relationship between Self-Attention and Convolutional Layers[C]// ICLR 2020.
>
> [3] Tsai Y H H, et al. Transformer Dissection: An Unified Understanding for Transformer’s Attention via the Lens of Kernel[C]// EMNLP 2019: 4344-4353.
>
> [4] Park N, Kim S. How Do Vision Transformers Work?[C]// ICLR 2022.

---

> ### Author Response · Authors · 2025-11-22
>
> > **W2:** Concern about sensitivity to hyperparameters. The proposed method introduces a non-trivial number of hyperparameters, and its performance is likely sensitive to their careful tuning. Key hyperparameters include the choice of intermediate transformer layers, the initial values for the learnable curvatures, and the loss weighting factor α. The paper would be strengthened by a sensitivity analysis demonstrating the robustness of the results to variations in these critical settings. Without such an analysis, it is difficult to assess the true practicality and generalizability of the approach, as the reported strong performance might be contingent on a specific, finely-tuned configuration.
>
> **R:** Thanks for your suggestions. We add experiments to analyze the sensitivity of hyperparameters, including the choice of intermediate transformer layers $p_j$, the initial values for the learnable curvatures, and the loss weighting factor $\alpha$. We conduct experiments on the CIFAR-100 dataset by adjusting these hyperparameters. Results presented in Table 2 show that the performance remains relatively stable with varying hyperparameter values, indicating that the method is only minimally sensitive to hyperparameter variation. We include these analyses in the revised manuscript.
>
> **Table 2(a): Sensitivity analysis on the choice of intermediate transformer layers.**
> | chosen intermediate layers |     LA    |    HCA    |    MTA    |
> |:--------------------------:|:---------:|:---------:|:---------:|
> |          [0,5,11]          |   71.08   |   52.52   |   84.65   |
> |          [2,6,11]          |   71.20   |   52.69   |   84.89   |
> |      [4, 7, 11] (ours)     | **71.37** | **53.19** | **85.29** |
> |         [6, 8, 11]         |   71.25   |   52.63   |   84.85   |
> |          [8,9,11]          |   71.18   |   52.46   |   84.67   |
>
> **Table 2(b): Sensitivity analysis on the initial value for the learnable curvatures.**
> | curv_init_image | curv_init_text |     LA    |    HCA    |    MTA    |
> |:---------------:|:--------------:|:---------:|:---------:|:---------:|
> |       0.25      |       0.5      |   70.62   |   52.70   |   84.95   |
> |       0.5       |      0.25      |   70.88   |   52.95   |   85.02   |
> |       0.5       |       0.5      | **71.37** | **53.19** | **85.29** |
> |       0.6       |       0.6      |   70.87   |   53.15   |   85.20   |
> |       0.7       |       0.7      |   71.18   |   52.72   |   85.04   |
>
> **Table 2\(c\): Sensitivity analysis on the loss weighting factor.**
> | $\alpha$  | LA        | HCA       | MTA       |
> |-----------|-----------|-----------|-----------|
> | 0.3       | 70.87     | 52.88     | 84.76     |
> | 0.4       | 71.02     | 52.90     | 84.90     |
> | 0.5(ours) | **71.37** | **53.19** | **85.29** |
> | 0.6       | 70.94     | 53.12     | 85.17     |
> | 0.7       | 70.74     | 52.94     | 84.99     |

---

> ### Comment · Reviewer_1GsE · 2025-11-27
>
> Thank you for the clarification. The rebuttal provides helpful intuition and additional experiments. Some concerns remain partially addressed. I think the use of information bottleneck still relies mainly on empirical choices rather than a stronger first-principle justification. That said, the overall contribution is clearer, and I will maintain my positive rating.

---

> > ### Author Response · Authors · 2025-11-27
> >
> > Thank you for your thoughtful review and positive feedback! We sincerely appreciate your valuable suggestions, which have been instrumental in enhancing this work!

---

### Official Review · Reviewer_jhgh · 2025-10-31

**Soundness:** 3
**Presentation:** 3
**Contribution:** 3
**Rating:** 6
**Confidence:** 4

**Summary:**

This paper addresses a key limitation in current vision-language models (VLMs): the asymmetric and potentially suboptimal alignment of modalities, where images are represented by single features while text uses multi-level features. The proposed method, Alignment across Trees, constructs hierarchical (tree-structured) features for both modalities and embeds them in separate hyperbolic manifolds with learnable curvatures. It further introduces a semantic-aware visual feature extraction framework based on cross-attention guided by textual prompts, and a heterogeneous manifold alignment technique that uses KL divergence between distributions on different manifolds via an intermediate curvature. Experiments on multiple taxonomic open-set classification benchmarks show that this approach outperforms existing methods in aligning hierarchical multimodal data.

**Strengths:**

S1. The paper identifies and tackles a issue in VLMs—the mismatch in representational hierarchy between vision and language. Figure 1 clearly illustrates this contrast, highlighting how the proposed tree-based structure achieves more symmetric alignment.

S2.This paper has comprehensive ablation and visualization.

**Weaknesses:**

W1. The Taylor approximation used to compute the KL-based manifold distance (Appendix A) lacks empirical evaluation. No sensitivity analysis is provided for the approximation constant 𝑟, raising concerns about robustness and stability.

W2. The paper does not analyze potential failure modes, computational overhead from learning multiple curvatures.

W3. The approach to class imbalance across different tree levels is not explained, leaving open questions about potential bias in curvature learning or alignment.

**Questions:**

Could visualizations be supported by metrics like inter/intra-class variance or cluster separation?

---

> ### Author Response · Authors · 2025-11-22
>
> Thanks for your insightful and thorough review. We will address your concerns one by one.
>
> > **W1:** The Taylor approximation used to compute the KL-based manifold distance (Appendix A) lacks empirical evaluation. No sensitivity analysis is provided for the approximation constant 𝑟, raising concerns about robustness and stability.
>
> **R:** Thanks for your suggestions. The Taylor approximation used to compute the KL-based manifold distance is applied on the hyperbolic distance. We conduct an empirical analysis to evaluate this approximation error in Taylor. Specifically, we randomly select 1,000,000 pairs of hyperbolic vectors generated by our method, compute both the exact hyperbolic distance and its Taylor-approximated value, and measure the relative error defined as $error = \frac{|d_{exact} - d_{approx}|}{|d_{exact}|}$.
> We plot the histogram of this ratio presented in Figure 7 in the Appendix C.7. Results show that the mean relative error is 5.3%, with a maximum of 32.4% and a minimum below 0.1%, demonstrating that the approximation error is acceptable and controllable in practice.
>
> The approximation constant r is the norm of feature midpoints on tangent spaces. In our implementation, r is estimated per batch by computing the mean feature vector on the tangent space and taking its norm. This estimation makes it naturally consistent with the training dynamics—its value evolves smoothly with the feature distribution of the model. We compute the relative error defined as $error_{r} = \frac{|r_{exact} - r|}{|r_{exact}|}$, where $r_{exact}$ denotes the exact constant that is computed on full-batch data. We further visualize the trajectory of $error_{r}$ during training (shown in Appendix C.8) and observe only minor fluctuations, confirming its empirical stability. Finally, we set r to a fixed constant, i.e., r=1 and r=2, where results are shown in Table 1. Results demonstrate that our method is robust to the variation of r. In conclusion, our method is robust to the approximation of r.
>
> We have included these analyses in the revised manuscript.
>
> **Table 1: Analysis on the robustness of r.**
> | r    | LA        | HCA       | MTA       |
> |------|-----------|-----------|-----------|
> | 1    | 70.58     | 52.41     | 84.48     |
> | 2    | 70.52     | 52.46     | 84.50     |
> | ours | **71.37** | **53.19** | **85.29** |
>
>
>
> > **W2:** The paper does not analyze potential failure modes, computational overhead from learning multiple curvatures.
>
>
> **R:** Thank you for pointing this out. Our method does not explicitly account for the fact that hierarchical data in real-world scenarios often exhibit domain shifts. As a result, our performance may be limited under cross-domain settings. In future work, we plan to develop a robust method that can handle such cross-domain variations, improving the generalization in the open world. We will add this to the Conclusion section in the revised manuscript.
>
> We further measure the computational overhead of our method and that of learning a single shared curvature in the Rare Species 1-shot setting with a batch size of 8. The time cost for the single-curvature method and our method is 74 s and 74.5 s per batch, and the memory cost is 10,400 MB and 10,400 MB, respectively. The results demonstrate that the additional time and memory overhead introduced by learning multiple curvatures is minimal and does not affect the overall efficiency. We include detailed analysis in the revised manuscript.
>
> > **W3:** The approach to class imbalance across different tree levels is not explained, leaving open questions about potential bias in curvature learning or alignment.
>
> **R:** Class imbalance is a common issue. CLIP has already shown the capability to handle such an imbalance issue [1]. Therefore, we did not explicitly consider the imbalance issue in our current design. Thank you for the suggestion. In future work, we plan to develop more robust algorithms to address this issue. As a preliminary attempt, we have implemented a sample re-weighting strategy to mitigate potential bias. As shown in Table 1, this balanced approach brings consistent improvements in hierarchical metrics (HCA and MTA). We attribute these improvements to the fact that the re-weighting strategy effectively addresses the imbalance at non-leaf nodes, allowing the model to better learn the hierarchical structure of the data.
>
> **Table 2: Analysis on class imbalance using MaPLe.**
> |                |   LA      |    HCA    |  MTA      |
> |----------------|-----------|-----------|-----------|
> | MaPLe          |   68.75   |    4.65   |   50.60   |
> | +ProTeCt       |   69.33   |   48.10   |   83.36   |
> | +Ours          | **71.37** |   53.19   | 85.29     |
> | +Ours-balanced |   71.26   | **53.54** | **85.70** |
>
> [1] Wen X, et al. What makes clip more robust to long-tailed pre-training data? a controlled study for transferable insights[J]. NeurIPS, 2024, 37: 36567-36601.

---

> ### Author Response · Authors · 2025-11-22
>
> > **Q:** Could visualizations be supported by metrics like inter/intra-class variance or cluster separation?
>
>
> **A:** Thanks for your suggestions. We compare inter/intra-class variance and cluster separation in our method against that of the baseline method (protect Wu et al. (2024)). Results in Table 2 show the superiority of our method.
>
> **Table 3(a): Silhouette Score(↑) of learned image representations.**
> |          |   Phylum   |   Class   |    Order   |   Family   |    Genus   |   Species  |
> |:--------:|:----------:|:---------:|:----------:|:----------:|:----------:|:----------:|
> | Baseline |   0.0106   |   0.0191  |   0.0301   |   0.0563   |    0.068   |   0.0621   |
> |   Ours   | **0.0112** | **0.025** | **0.0381** | **0.0671** | **0.0796** | **0.0727** |
>
> **Table 3(b): Calinski-Harabasz Index(↑) of learned image representations.**
> |          |   Phylum  |   Class   |   Order  |  Family  |   Genus  |  Species |
> |:--------:|:---------:|:---------:|:--------:|:--------:|:--------:|:--------:|
> | Baseline |   105.7   |   113.1   |   48.5   |   33.8   |   27.0   |   23.4   |
> |   Ours   | **128.0** | **136.4** | **56.8** | **39.5** | **31.6** | **27.3** |
>
> **Table 3\(c\): Davies-Bouldin Index(↓) of learned image representations.**
> |          |  Phylum  |  Class   |   Order  |  Family  |   Genus  |  Species |
> |:--------:|:--------:|:--------:|:--------:|:--------:|:--------:|:--------:|
> | Baseline |   4.48   |   3.92   |   3.33   |   3.24   |   3.16   |   3.38   |
> |   Ours   | **3.78** | **3.50** | **3.04** | **2.99** | **2.94** | **3.19** |

---

### Official Review · Reviewer_L9Aq · 2025-11-03

**Soundness:** 3
**Presentation:** 2
**Contribution:** 3
**Rating:** 4
**Confidence:** 3

**Summary:**

This paper proposes a hyperbolic representation approach for vision–language alignment in a hierarchical manner.

The approach assumes we have textual labels at multiple levels (e.g. not just the original animal species but also the biological family, order, and class). It hypothesizes that different layers of a deep model similarly discern representations at different levels of granularity that can be aligned to the different textual hierarchy levels.

Based on this idea, the paper investigates hyberbolic representations. First, hierarchical representations are induced for the two modalities in a deep model by eliminating cross-token self-attention, adding cross-modality attention. The two modality-specific manifolds are bridged by constructing an intermediate hyperbolic space, to which the original ones are mapped subject to hierarchical entailment constraints.

**Strengths:**

- Interesting idea worth exploring
- Some promising experimental results
- Generalization to novel classes investigated

**Weaknesses:**

- Most of the examples focus on hierarchical relationships from biological taxonomy. While there are experiments on more diverse datasets such as ImageNet, the paper does not provide much analysis and insight into how the model behaves on other kinds of hierarchies, especially when the data is more diverse.

- On the Rare Species Dataset, the authors do not stick with the original labels but instead use hierarchical label names created by concatenating the various hierarchy levels in Algorithm 1 (Hierarchical Tree Construction). This seems to massively alter the nature of the task, as now the parent hierarchy is already given in each textual label. If the motivation for this step, as claimed by the authors, is just to deal with ambiguous labels, then why not add a small extra term just to ambiguous labels?

- The paper currently lacks a larger discussion of methods for hierarchy/taxonomy alignment. The paper presents a single method but doesn't explain the design space and design choices well enough. What alternatives could have been chosen? The relationship to the broader field of hierarchical/taxonomy alignment could be explored further.

- The alignment occurs in multiple phases rather than a single one, in particular it occurs via the Heterogeneous Manifold Alignment Algorithm but also earlier via cross-attention in the Semantic-Aware Visual Feature Extraction Framework. This cross-attention between image and text in the Semantic-Aware Visual Feature Extraction Framework (Eq. 5) appears to allow information to be copied over between modalities. It is not obvious how the approach ensures that each modality indeed yields a hierarchy for that particular modality rather than just copying over information from the other modality. How well does the model work without this cross-attention?

**Questions:**

- Understanding how the model avoids degenerate solutions is important. Can you explain to readers of the paper which parts of the model ensure that the fine-grained features do not become overly coarse?

- What do we observe on truly diverse classes, e.g. WordNet's "artifact" class?

Minor issues:

- The explanation of Eq. 3 mistakenly refers to $\delta$ and $u$ that do not appear in the formula.

- bad formatting in L. 197 "fine-grained information(Chen et al., 2024)."

- typo "can not"

- The paper suffers from extensive negative vspace hacking. There are various orphan headings such as "4.2 HETEROGENEOUS MANIFOLD ALIGNMENT ALGORITHM" and "4.2.2 INTER-MODAL GEOMETRIC ALIGNMENT MECHANISM". There are also many headings that have insufficient spacing, e.g. "4.2.1 INTERMEDIATE MANIFOLD CONSTRUCTION" and "4.4 OPTIMIZATION STRATEGY".

- There are a number of poorly typeset mathematical expressions, e.g. Eq. (49).

---

> ### Author Response · Authors · 2025-11-22
>
> Thanks for your insightful and thorough review. We will address your concerns one by one.
>
> > **W1:** Most of the examples focus on hierarchical relationships from biological taxonomy. While there are experiments on more diverse datasets such as ImageNet, the paper does not provide much analysis and insight into how the model behaves on other kinds of hierarchies, especially when the data is more diverse.
> **Q2:** What do we observe on truly diverse classes, e.g. WordNet's "artifact" class?
>
>
> **R:** Thanks for your suggestions. We additionally conduct visualization analyses on artifact classes from the ImageNet dataset, including representative examples such as taxicabs, police vans, and carousels. The results, presented in Figures 10 and 11 in the updated Appendix C.12, demonstrate that our method successfully captures coarse-to-fine semantic granularities on these diverse artifact categories as well.

---

> ### Author Response · Authors · 2025-11-22
>
> > **W2**: On the Rare Species Dataset, the authors do not stick with the original labels but instead use hierarchical label names created by concatenating the various hierarchy levels in Algorithm 1 (Hierarchical Tree Construction). This seems to massively alter the nature of the task, as now the parent hierarchy is already given in each textual label. If the motivation for this step, as claimed by the authors, is just to deal with ambiguous labels, then why not add a small extra term just to ambiguous labels?
>
>
>
>
> **R:** We use concatenated hierarchical labels in the Rare Species Dataset to address its ambiguous label issue, following the baseline method BioCLIP (Stevens et al., 2024). Other datasets do not have this issue, so we do not apply concatenation there.
>
> Thanks for pointing it out. We consider an alternative approach to handle ambiguity. Specifically, we replace non-unique labels with a combination of the scientific name and the common name to guarantee uniqueness without relying on the concatenated hierarchy. We re-conduct experiments without using the concatenation strategy. Results shown in Table 1 show that our method significantly improves the performance of the compared methods, demonstrating its superiority. Notably, in the 16-shot setting of the few-shot task, our method achieves up to a 22.03% improvement on LA, a 27.82% improvement on HCA, and a 15.58% improvement on MTA, indicating its ability to align the textual and visual modalities with the hierarchical semantic structures. These additional experimental results have been included in Appendix C.9 of the revised manuscript.
>
>
>  **Table 1(a): 1-shot TOS classification using MaPLe**
> |            |  LA |    HCA    |  MTA  |
> |:----------:|:---------:|:---------:|:---------:|
> |   MaPLe    | 38.09     | 1.84      | 35.19     |
> |  +ProTeCt  | 35.67     | 7.84      | 61.69     |
> |    +Ours   | **38.34** | **10.55** | **69.47** |
>
> **Table 1(b): 1-shot TOS classification using PromptSRC**
> |            |  LA       |    HCA    |  MTA      |
> |:----------:|:---------:|:---------:|:---------:|
> |  Promptsrc | 39.13     | 6.09      | 43.68     |
> |  +ProTeCt  | 37.92     | 9.93      | 64.45     |
> |    +Ours   | **39.97** | **12.47** | **71.93** |
>
> **Table 1\(c\): 16-shot TOS classification using MaPLe**
> |            |  LA |    HCA    |  MTA  |
> |:----------:|:---------:|:---------:|:---------:|
> |   MaPLe    | 50.69     | 2.01      | 35.19     |
> |  +ProTeCt  | 44.68     | 16.19     | 61.69     |
> |    +Ours   | **66.71** | **44.01** | **77.27** |
>
> **Table 1(d): 16-shot TOS classification using PromptSRC**
> |            | LA |  HCA  | MTA |
> |:----------:|:--------:|:-----:|:-------:|
> |   PromptSRC  | 59.99    | 4.55  | 40.54   |
> |  +ProTeCt  | 55.53    | 28.66 | 74.76   |
> |    +Ours   | **65.16**    | **44.43** | **79.74**   |
>
> **Table 1(e): base-to-base/base-to-novel/base-to-whole generalization using MaPLe**
> |            |   LA-Base  |  LA-Novel  |    LA-HM   | LA-Whole   |  HCA-Base  |  HCA-Novel |   HCA-HM   | HCA-Whole  |  MTA-Base |  MTA-Novel |   MTA-HM   | MTA-Whole  |
> |:----------:|:----------:|:----------:|:----------:|------------|:----------:|:----------:|:----------:|------------|:---------:|:----------:|:----------:|------------|
> |   MaPLe  | 50.38      | 39.29      | 44.15      | 41.09      | 3.20       | 2.90       | 3.04       | 1.84       | 8.51      | 8.15       | 8.33       | 7.42       |
> |  +ProTeCt  | 45.50      | 38.80      | 41.88      | 37.92      | 21.78      | 8.44       | 12.17      | 10.97      | 18.69     | 17.45      | 18.05      | 18.14      |
> |    +Ours   | **66.86** | **40.40** | **50.37** | **47.77** | **50.97** | **10.10** | **16.86** | **24.57** | **19.15** | **20.01** | **19.57** | **20.07** |
>
>
> **Table 1(f): base-to-base/base-to-novel/base-to-whole generalization using PromptSRC**
> |            |  LA-Base  |  LA-Novel |   LA-HM   | LA-Whole  |  HCA-Base | HCA-Novel |   HCA-HM  | HCA-Whole |  MTA-Base | MTA-Novel |   MTA-HM  | MTA-Whole |
> |:----------:|:---------:|:---------:|:---------:|-----------|:---------:|:---------:|:---------:|-----------|:---------:|:---------:|:---------:|-----------|
> |   PromptSRC  | 60.13     | 42.20     | 49.59     | 46.43     | 6.06      | 6.04      | 6.05      | 4.01      | 7.64      | 8.00      | 7.82      | 6.25      |
> |  +ProTeCt  | 54.33     | 42.02     | 47.39     | 44.93     | 33.22     | 11.75     | 17.36     | 17.36     | 18.08     | 16.12     | 17.04     | 17.88     |
> |    +Ours   | **64.93** | **44.21** | **52.60** | **48.85** | **48.36** | **12.09** | **19.34** | **24.74** | **18.92** | **16.72** | **17.75** | **18.42** |

---

> ### Author Response · Authors · 2025-11-22
>
> > **W3:** The paper currently lacks a larger discussion of methods for hierarchy/taxonomy alignment. The paper presents a single method but doesn't explain the design space and design choices well enough. What alternatives could have been chosen? The relationship to the broader field of hierarchical/taxonomy alignment could be explored further.
>
> **R:** Thanks for your suggestions. We have added a broader discussion on the taxonomy classification task and clarified why we choose symmetric alignment and hyperbolic manifolds in the Related Work section. We have also added a discussion on the alternative methods to the Future Work section.
>
>
> 1. Added Related Work
>
> Taxonomy classification aims to predict labels at different levels of a class hierarchy. Several works (Goo et al., 2016; Kim et al., 2018) utilize CNN-based networks to learn a hierarchical feature space. These methods rely on predefined hierarchical structures and fixed sets of categories for classification. To solve this issue, some works construct a multi-modal alignment method for taxonomy classification. Wu et al. (2024) first extract a single visual feature and compute contrastive losses with multi-level textual features using prompt learning, introducing metrics for hierarchical consistency. BioCLIP (Stevens et al., 2024), BioCLIP2 (Gu et al., 2025), and Biotrove (Yang et al., 2024) form prompts from coarse-to-fine annotations for pretraining. Sastry et al. apply transitive entailment constraints to features of hierarchical textual labels. However, these methods align hierarchical textual features with a single visual representation (i.e., asymmetric alignment) on the Euclidean spaces, leading to **suboptimal alignment** and ignoring the geometric structure of hierarchical multimodal data. In contrast, our method extracts hierarchical visual features to align hierarchical textual features (i.e., symmetric alignment), and we model them in hyperbolic manifolds for improved alignment. The reason we choose hyperbolic manifolds is that they offer a more effective way to capture hierarchical geometric structures, as their volume grows exponentially with the radius, aligning with the exponential increase in data size along the depth of hierarchical structures (Fan et al., 2025b).
>
> Goo W, et al. Taxonomy-regularized semantic deep convolutional neural networks[C]//ECCV. Cham: Springer International Publishing, 2016: 86-101.
>
> Kim H J, Frahm J M. Hierarchy of alternating specialists for scene recognition[C]//ECCV. 2018: 451-467.
>
> Gu J, et al. Bioclip 2: Emergent properties from scaling hierarchical contrastive learning[C]. NeurIPS, 2025.
>
> Yang C H, et al. Biotrove: A large curated image dataset enabling ai for biodiversity[C]. NeurIPS, 2024, 37: 102101-102120.
>
> Sastry S, et al. Global and Local Entailment Learning for Natural World Imagery[C]. ICCV, 2025.
>
> Fan X et al. Curvature Learning for Generalization of Hyperbolic Neural Networks[J]. IJCV, 2025: 1-37.
>
> 2. Added Future Work
>
> Considering the complexity of the underlying geometric structures, a single-curvature hyperbolic manifold is inherently limited in its ability to fully capture such rich semantics. In future work, we plan to leverage more expressive mixed-curvature spaces to separately model visual and textual features.

---

> ### Author Response · Authors · 2025-11-22
>
> > **W4:** The alignment occurs in multiple phases rather than a single one, in particular it occurs via the Heterogeneous Manifold Alignment Algorithm but also earlier via cross-attention in the Semantic-Aware Visual Feature Extraction Framework. This cross-attention between image and text in the Semantic-Aware Visual Feature Extraction Framework (Eq. 5) appears to allow information to be copied over between modalities. It is not obvious how the approach ensures that each modality indeed yields a hierarchy for that particular modality rather than just copying over information from the other modality. How well does the model work without this cross-attention?
>
>
> **R:** Cross-attention mechanism in the Semantic-Aware Visual Feature Extraction Framework is utilized to compute the weights of the intermediate visual tokens to obtain hierarchical visual features, rather than copying over between modalities. Concretely, textual features are used as queries to calculate the weights of intermediate visual tokens. This weighting process helps guide the extraction of hierarchical visual features at the corresponding granularity. Importantly, the final visual features are the weighted sum of visual modality tokens. Thus, the text serves only as a selector to influence the weighting, without directly affecting, copying, or replacing the visual features.
>
> We conduct an ablation analysis on the cross-attention mechanism by removing it. Specifically, we manually design a set of weighting coefficients for the CLS tokens at different layers to construct hierarchical visual features. For example, the weight for level 1 is [0.2397, 0.3235, 0.4368], and the weight for level 2 is [0.1628，0.2966, 0.54051] for training. During inference, given an image and a set of candidate textual labels, we compute visual features at all levels and evaluate their similarities to all textual labels, and the label with the highest similarity is the final prediction. Results are shown in Table 2, highlighting the importance of this mechanism.
>
> **Table 2(a): Analysis on cross attention module using MaPLe.**
> |                            |  LA |    HCA    |  MTA  |
> |:--------------------------:|:---------:|:---------:|:---------:|
> |           MaPLe          |   68.75   |    4.65   |   50.60   |
> |          +ProTeCt          |   69.33   |   48.10   |   83.36   |
> |            +Ours           | **71.37** | **53.19** | **85.29** |
> | +Ours w/o  cross attention |   67.43   |   48.42   |   84.04   |
>
> **Table 2(b): Analysis on cross attention module using PromptSRC.**
> |                            |  LA |    HCA    |  MTA  |
> |:--------------------------:|:---------:|:---------:|:---------:|
> |           PromptSRC          |   72.48   |   14.36   |   51.91   |
> |          +ProTeCt          |   73.07   |   49.54   |   85.16   |
> |            +Ours           | **73.54** | **51.91** | **85.76** |
> | +Ours w/o  cross attention |   73.12   |   50.11   |   85.08   |
>
>
> > **Q1:** Understanding how the model avoids degenerate solutions is important. Can you explain to readers of the paper which parts of the model ensure that the fine-grained features do not become overly coarse?
>
> **A:** Our model avoids the degeneration of fine-grained features through three mechanisms.
> - Feature construction preserves fine-level detail. We include the final-layer class token to construct visual representations. We utilize the finest-grained textual feature as the query to compute the attention weights on candidate tokens. Average attention weights on layers 4, 7, and 11 are 0.06, 0.08, and 0.86, respectively. We can observe that attention weights on this final-layer token are significantly larger than token on intermediate layers. This indicates that the finest-grained features include fine-grained details and do not collapse into coarser ones.
> - Fine-grained alignment loss directly prevents collapse. Our training objective $J_{\text{pro}}(T_e, V_e)$ in Eq. (17) explicitly includes a distance alignment loss between finest-level textual and visual features, which forces the model to maintain discriminative fine-grained structure, since any degeneration would increase this loss.
> -  Modality-specific geometric constraints in Eq. (16) enforce boundary separation. The constraints in each modality encourage fine-grained features to reside near the boundary of hyperbolic manifolds. Because boundary points have larger radii and correspond to more specific semantics, our method prevents fine-level collapsing to coarser levels.

---

> ### Author Response · Authors · 2025-11-22
>
> > **minor issue 1:** The explanation of Eq.(3) mistakenly refers to $\delta$ and $\mu$ that do not appear in the formula.
>
> **R:** Thank you for the careful review. We apologize for the notation inconsistency in Eq. (3). We have corrected the formula to properly correspond with the parameters $\boldsymbol{u}$ and $\delta$ described in the text. The corrected equation is:
>
> $$
> \mathcal{N}\_{\mathcal{L}^{c}}(\boldsymbol{x} \mid \boldsymbol{u}, \delta) = \frac{1}{Z(\delta)}\exp\left(-\frac{d^2\_{c}(\boldsymbol{x}, \boldsymbol{u})}{2\delta^2}\right).
> $$
>
> We have updated this in the revised manuscript.
>
>
> > **minor issues 2&3:** bad formatting in L. 197 "fine-grained information(Chen et al., 2024)."; typo "can not".
>
> **R:** Thank you for noting these issues. We have fixed the formatting by adding a space before the citation and corrected "can not" to "cannot" throughout the manuscript.
>
> > **minor issue 4:** The paper suffers from extensive negative vspace hacking. There are various orphan headings such as "4.2 HETEROGENEOUS MANIFOLD ALIGNMENT ALGORITHM" and "4.2.2 INTER-MODAL GEOMETRIC ALIGNMENT MECHANISM". There are also many headings that have insufficient spacing, e.g. "4.2.1 INTERMEDIATE MANIFOLD CONSTRUCTION" and "4.4 OPTIMIZATION STRATEGY".
>
> **R:** We appreciate you bringing this to our attention. We have carefully reviewed and adjusted all spacing throughout the manuscript, ensuring proper spacing between sections. All orphan headings have been addressed, and we have verified that each heading now has sufficient spacing above and below.
>
> > **minor issue 5:** There are a number of poorly typeset mathematical expressions, e.g. Eq. (49).
>
> **R:** Thank you for this observation. We have carefully reviewed and corrected the typesetting of all mathematical expressions in the manuscript. For example, Eq. (59) has been reformatted as:$\mathit{Metric}\_{\mathrm{harmony}} = \frac{2 \cdot \mathit{Metric}\_{\mathrm{base}} \cdot \mathit{Metric}\_{\mathrm{novel}}}{\mathit{Metric}\_{\mathrm{base}} + \mathit{Metric}\_{\mathrm{novel}}}$

---

### Author Response · Authors · 2025-12-02

# Final Remarks
Dear Area Chair,

We sincerely thank you for your time and effort in handling our submission. We are truly encouraged by the positive assessments and constructive feedback from all reviewers, which have helped us significantly refine our manuscript. We especially appreciate your support in facilitating these productive discussions.

**Summary of Recognized Strengths**: We are grateful for the reviewers’ recognition of our work and its key contributions:
- Identification of the key limitation of feature representation asymmetry in VLMs (jhgh, 1GsE, 8Seg)
- Interesting and novel approach leveraging hyperbolic geometry (L9Aq, 1GsE, 8Seg)
- Comprehensive experiments, ablation studies, and visualizations (jhgh, 1GsE)
- Strong theoretical foundation supported by proofs (8Seg)
- Strong performance gains and generalization capabilities (All reviewers)

**Summary of the Main Concerns Raised by Reviewers and Our Rebuttal Efforts**: We thank the reviewers for pointing out the concerns that help us improve our work:

>**Visualizations**: Requests to provide attention maps for the baseline, extend visualizations to more diverse classes, and complement learned representations with quantitative metrics (L9Aq, jhgh, 8Seg).

- **Response:** We add attention map comparisons against the baseline and extend visualizations to ImageNet, demonstrating our method's ability to adaptively attend to hierarchically relevant regions for diverse classes. Additionally, we compute quantitative metrics (Silhouette Score, Calinski-Harabasz Index, and Davies-Bouldin Index), which verify that our method produces significantly more discriminative feature representations across all hierarchical levels.


>**Related Work**: Suggestions to expand the discussion on hierarchy/taxonomy alignment, relative representations, and identifiability in representation learning (L9Aq, 8Seg).

- **Response:** We add the discussion on taxonomy alignment, relative representations, and identifiability in representation learning to the related work section.



>**Sensitivity Analysis**: Concerns regarding the lack of sensitivity analysis for both hyperparameters and model architecture (1GsE).


- **Response:** We analyze the sensitivity of the weighting factor $\alpha$, the selection of intermediate layers, and backbones with varying ViT depths, demonstrating the robustness of our method to variations in both hyperparameters and model architectures.



>**Ablation Studies**: Requests for additional ablations on specific components, including the manifold distance design and the cross-attention module (8Seg, L9Aq).

- **Response:** We compare our method with three alternative distance designs, demonstrating its advantage in capturing the underlying geometry. We further conduct an ablation by substituting the cross-attention module with manually specified weights; the resulting performance drop demonstrates the necessity of the cross-attention design.



>**Additional Analysis**: Suggestions to analyze Taylor approximation errors, computational overhead, and learned curvatures (jhgh, 8Seg).

- **Response:** We empirically quantify the Taylor approximation error by comparing the exact distances with their Taylor estimates, finding the deviation to be negligible. Additionally, our efficiency analysis demonstrates that the introduction of learnable curvatures imposes minimal computational overhead. Furthermore, we analyze the learned curvatures across datasets, showing that both the learned curvatures of different datasets and those of different modalities vary substantially, thereby demonstrating the necessity of our approach.


>**Discussions and Justifications**: Requests for theoretical justification of the semantic-aware feature extraction module, discussions on potential data imbalance across levels, and clarification on how the model avoids degenerate solutions (1GsE, jhgh, L9Aq).

- **Response:** We justify the selection of intermediate layers using the Information Bottleneck (IB) theory and empirically verify the method's robustness to specific layer choices. Additionally, we demonstrate our method's effectiveness under class imbalance. We further clarify that our model avoids degenerate solutions by feature construction, fine-grained alignment loss, and modality-specific geometric constraints.

---

> ### Author Response · Authors · 2025-12-02
>
> **Summary of Revisions for Camera-Ready**: To further improve the quality of our manuscript, we have integrated the rebuttal updates into the revised paper. Specifically, we:
>
> - **Incorporated the new experimental results**, including the sensitivity analyses, ablation studies, and comprehensive visualizations listed above.
> - **Integrated the expanded discussions and analyses** into the main text and appendix to provide clearer theoretical and empirical support.
> - **Expanded the Conclusion section** to explicitly discuss the limitations of our current approach and outline potential directions for future work.
> - **Corrected minor typographical errors** and formatting inconsistencies in equations and text to ensure a polished presentation.
>
> Thank you again for your valuable contributions to strengthening this work.
>
> Best regards,
>
> Authors

---

### Meta-Review · Area_Chair_xtyr · 2026-01-07

**Summary:**

This paper addresses the asymmetric modality alignment problem in vision-language models, where existing methods align hierarchical textual features with only a single visual feature representation. Experiments on taxonomic open-set classification tasks across four datasets demonstrate consistent improvements.

The reviewers recognized several strengths of the work, including the clear identification of the limitation in VLMs. The reviewers also praised the novel use of heterogeneous hyperbolic manifolds with learnable curvatures. Reviewers also raised concerns about the limited scope of evaluation focusing on taxonomic classification. They are also concerned  about the practical impact of Taylor approximation errors.

Overall, the idea of constructing symmetric hierarchical features for both modalities and aligning them on hyperbolic manifolds represents a meaningful contribution to vision-language alignment research. And the experimental results consistently demonstrate the method's effectiveness across multiple datasets and evaluation protocols.

**Reviewer Concerns:**

Addressed concerns:
Visualization quality. The authors added attention map comparisons against the ProTeCt baseline, demonstrating that their method adaptively attends to hierarchically relevant regions.
Related works. The authors expanded discussion of relative representations.
Sensitivity analysis. The authors conducted experiments with varying intermediate layers and the loss weighting factor.,

Outstanding concerns:
The remaining concern is the limited evaluation scope, as all experiments focus on taxonomic open-set classification. While the authors showed results on ImageNet artifact classes, this still falls within the taxonomic classification framework rather than demonstrating broader applicability.

**Reviewer Scores:**

Reviewer L9Aq initially rated the paper 4, marginally below acceptance threshold. Based on the rebuttal addressing concerns about artifact classes and providing an alternative labeling strategy, this reviewer would likely increase their score to 5/6. The visualization of attention maps on diverse ImageNet classes and the experiments showing consistent improvements even without the concatenation strategy partially address concerns about generalizability beyond biological taxonomy. However, the fundamental question about how the method behaves on truly diverse hierarchical relationships beyond the evaluated datasets remains incompletely addressed.

---

### Decision · Program_Chairs · 2026-01-26

Accept (Poster)